# Economy-wide evaluation of $CO_2$ and air quality impacts of electrification in the United States

John E. T. Bistline [1] ✉, Geoffrey Blanford[1], John Grant[2], Eladio Knipping[1], David L. McCollum [3], Uarporn Nopmongcol[2], Heidi Scarth [1], Tejas Shah[2] & Greg Yarwood [2]

Adopting electric end-use technologies instead of fossil-fueled alternatives, known as electrification, is an important economy-wide decarbonization strategy that also reduces criteria pollutant emissions and improves air quality. In this study, we evaluate $CO_2$ and air quality co-benefits of electrification scenarios by linking a detailed energy systems model and a full-form photochemical air quality model in the United States. We find that electrification can substantially lower $CO_2$ and improve air quality and that decarbonization policy can amplify these trends, which yield immediate and localized benefits. In particular, transport electrification can improve ozone and fine particulate matter ($PM_{2.5}$), though the magnitude of changes varies regionally. However, growing activity from non-energy-related $PM_{2.5}$ sources—such as fugitive dust and agricultural emissions—can offset electrification benefits, suggesting that additional measures beyond $CO_2$ policy and electrification are needed to meet air quality goals. We illustrate how commonly used marginal emissions approaches systematically underestimate reductions from electrification.

Adopting electric end-use technologies in place of fossil-fueled alternatives, known as electrification (in the developed world context), has been shown to be an important element of economy-wide decarbonization efforts[1–3]. Reducing power sector emissions and then using that low-emitting electricity to reduce end-use emissions is employed alongside energy efficiency, carbon capture and removal, other low-carbon fuels, and demand-side responses (including the lowering of energy demand through structural and societal change) to decarbonize energy systems and address climate change. Increased adoption of electric end-use technologies across buildings, industrial, and transport sectors is driven by a combination of technological change, consumer choice, and policy, especially those targeting $CO_2$ reductions[4].

In addition to lowering $CO_2$ emissions from fuel combustion, electrification also can reduce criteria pollutants and improve air quality[5]. Currently, the transportation and electricity generation sectors are leading emitters of U.S. $CO_2$ and also criteria pollutants, including oxides of nitrogen ($NO_x$) and oxides of sulfur ($SO_x$). Their emissions contribute to the formation of common and widespread air pollutants, such as ozone and fine particulate matter (PM). PM can be directly emitted from these sources (primary) or formed from gaseous precursors (secondary). Research suggests that monetized benefits of climate change mitigation are high for air quality improvements, which have immediate and localized benefits[6]. However, these impacts are uncertain due to unknowns about the speed and scale of electrification, the complexity of interactions between emissions and air quality, as well as the fact that other sources of emissions, such as fugitive dust, agriculture, and solvent use, also affect air quality (fugitive dust includes emissions from anthropogenic disturbances, such as resuspension of road dust, agricultural dust, and construction dust, whereas natural dust is from wind disturbances on land surfaces). These changes have large implications for human health, for policymakers

[1]Electric Power Research Institute, 3420 Hillview Avenue, Palo Alto, CA 94304, USA. [2]Ramboll, 7250 Redwood Blvd., Suite 105, Novato, CA 94945, USA. [3]Oak Ridge National Laboratory, 2360 Cherahala Blvd, Knoxville, TN 37932, USA. ✉e-mail: jbistline@epri.com

deciding between approaches for managing $CO_2$ emissions and air quality, and for energy system planners.

Despite these high stakes, few studies have examined air quality impacts of economy-wide electrification or drivers of air quality outcomes by linking a detailed energy systems model with a full-form photochemical air quality model. Previous studies examine emission changes for $NO_x$ and other criteria pollutants due to vehicle electrification in the U.S.[7–11]. Many analyses address potential consequences of electrification in the U.S. energy system using optimization models[12–16], but do not pursue the next step past emissions modeling. Connections between emission changes due to electrification and air concentrations have been made through life cycle assessment (LCA) models accounting for upstream fuel-sector emissions[17]. However, LCA models are not readily able to simulate important dynamics across source sectors. Other assessments of air quality impacts from mitigation use stylized emissions scenarios rather than detailed structural energy systems models, which can evaluate drivers and impacts of emissions changes[6,18,19]. Finally, emissions factors are widely applied to estimate potential emissions impacts from electrification, energy efficiency, and other interventions[10,20–22].

Uncertainties in future emissions from drivers such as mitigation policies, land-use change, and technology development also have been explored using global integrated assessment models (IAM)[23,24]. The resulting emissions inventories are used as inputs into air quality models (AQM), either chemical transport models, or reduced-form models, to evaluate pollutant concentrations[25]. Reduced-form models use simplified relationships between the inputs and outputs of a full physics AQM, so can diverge to varying degrees depending on the complex pattern of emission changes. IAMs themselves and other energy system models can have relatively coarse temporal[26] and spatial resolution[27,28], and use more aggregated source sectors (e.g., economic sectors) than standard emission inventories used in AQM (e.g., Source Classification Code, SCC, level). Bridging between IAM and AQM may neglect variability within a source sector[29,30]. Many comprehensive AQM studies adopt pre-specified market shares[31–36], but this approach neglects important cross-sector interactions such as how changes in electricity demand could influence electricity prices which in turn affects end-use adoption.

This study distinguishes itself from previous efforts in several ways. First, we used an integrated electric and energy systems model with temporal, spatial, sectoral, and technological detail. These features provide endogenous responsiveness of electricity demand and hourly electric load shapes to a range of scenario-specific drivers, including incentives for electrification and $CO_2$ policy, which improves upon previous efforts with exogenous emissions scenarios, end-use representations, and sectoral adoption sensitivities[6,32,37,38]. Second, we develop scenarios that present a range of potential futures with increasing degrees of electrification, including in response to decarbonization policy. We use a state-of-the-science full-form approach for air quality, i.e., simulating explicit photochemistry with hourly time resolution over a complete annual cycle. The full-form AQM is necessary to address the nonlinear chemistry of ozone and PM seen in this work due to drastically changing emission profiles, spatially and temporally, for distinct chemical species from electrification and decarbonization. The detailed emission inventories in this work promote both inter-sectoral linkages and within-sector variability. We incorporate existing air quality plans into our analysis and apply the established methods for assessing future air quality[39] so that we can draw conclusions about future compliance with U.S. National Ambient Air Quality Standards (NAAQS). This high policy granularity and ability to examine future compliance sets our analysis apart from earlier efforts. Unlike papers that focus on specific geographies[35], our paper links a detailed energy systems model with a full-form photochemical air quality model of the contiguous United States to examine scenarios

that present a range of potential futures with increasing degrees of electrification evaluated with an economy-wide model. Linking these models provides a more complete picture of energy system transformations and their localized air quality impacts. The U.S.-wide geographical scope provides a unique contribution in showing spatial variation in air quality responses to electrification and decarbonization.

## Results

### Modeling energy system and air quality impacts of electrification

The U.S. Regional Economy, GHG, and Energy (US-REGEN) framework combines a state-of-the-art electric sector capacity planning and dispatch model with a uniquely capable end-use model[40,41]. Distinguishing features of the model include: (1) Detailed disaggregation of end-use sectors, activities, end-uses, and technologies and explicit tracking of structural classes including building type and size, building and equipment vintage, household attributes, and annual temperature profile; (2) endogenous end-use technology adoption; and (3) synchronized equilibrium of hourly load profiles and prices between electricity supply and energy use. These features enable US-REGEN to systematically represent many important dimensions of end-use technology tradeoffs, such as the heterogeneity of applications and customers, which better captures economic, behavioral, and policy factors influencing electrification and consequently emissions. Moreover, the integrated representation of electricity supply and demand provides a dynamic and scenario-consistent treatment of the marginal emissions from increased electric generation to support electrification, taking into account structural changes to the generation mix over time rather than relying on a historical snapshot, as many marginal emissions studies of electrification do[20]. Note that the energy system model scope for this analysis does not include endogenous representations of fossil fuel prices, biofuel production, non-electric fuel movement, or general equilibrium effects. US-REGEN is documented in detail in EPRI (2020)[41], so only summaries of key features are provided in the Methods and Supplemental Information.

We consider three scenarios that differ in the extent of electrification and drivers of decarbonization:

- Limited electrification: This scenario assumes limited electric vehicle (EV) adoption for light-duty vehicles, no growth in building electrification, and no concerted federal climate policy. These assumptions lead electricity demand to be approximately flat over time. This hypothetical benchmark was constructed to display conservative levels of electrification that, when compared to results from other scenarios, help to better quantify the impact of electrification on outcomes of interest.

- High electrification without carbon price: This scenario includes more optimistic assumptions about advanced end-use technologies but does not include national $CO_2$ policies. Higher electrification in this scenario is driven by: (1) allowing EVs to deploy endogenously (given continuing trends of falling battery costs); (2) accelerating the performance assumptions for heat pumps; and 3) faster depreciation of the existing equipment stock. This scenario more closely approximates expected market trends than does the Limited Electrification case, which allows us to evaluate the effects of such electrification.

- High electrification with carbon price: This scenario has the same technology assumptions as the second scenario and introduces a carbon price starting in 2025 at \$50/t$CO_2$ (in 2020 USD) and growing at 7% per year (\$271/t$CO_2$ in 2050), which is intended as a proxy for a suite of $CO_2$ policies for the electric and end-use sectors.

Detailed scenario assumptions are provided in Supplementary Note 6.

The US-REGEN model estimates $CO_2$ and criteria air emissions from anthropogenic sources for the entire Continental U.S. (CONUS) for each scenario over the period 2015–2050 in five-year increments. We model air quality for four scenarios: (1) 2035 Limited Electrification; (2) 2035 High Electrification without Carbon Price; (3) 2035 High Electrification with Carbon Price; and (4) 2050 High Electrification with Carbon Price. Modeling these four scenarios provides a wide range of potential futures and probes the relative roles of electrification technology and carbon pricing to air quality. The criteria emission estimates are made for ozone and PM precursors, including $NO_x$, $SO_x$, volatile organic compounds (VOC), carbon monoxide (CO), ammonia ($NH_3$), and primary PM. These emissions are processed to develop more temporally, spatially, and chemically refined inputs for air quality modeling. US-REGEN emissions by sector, source, fuel type, and region are cross-referenced to Source Classification Codes (SCC), county, and North American Industry Classification System (NAICS) code (see Supplementary Note 2 for more information).

We use the Comprehensive Air Quality Model with Extensions (CAMx), version 7.0, for air quality modeling with a 12-km grid covering the entire lower 48 states and nested within a larger 36-km grid (Supplementary Fig. S4) for every hour of the calendar year 2016. The meteorology for 2016 is used for all scenarios so that the projected air quality changes can be attributed solely to emissions changes. Our simulations build from a CAMx database developed by EPA and used for national air quality policy[42].

## Energy system and emissions results

Results in US-REGEN show how electrification can significantly raise electricity demand (Fig. 1) and electricity's share of final energy (Supplementary Fig. 17). Electricity currently represents about 20% of final energy (similar levels to the Limited Electrification scenario in future years), which grows in the High Electrification scenario to 31% in 2035 and 34% in 2050 (and to 34 and 51% when $CO_2$ policy is added), driven by technological change and further bolstered by $CO_2$ policy. Electricity demand is roughly flat over time in the Limited Electrification scenario, as growing service demand is offset by efficiency increases. The High Electrification scenario entails 23% (39%) growth by 2035 (2050), which increases with $CO_2$ policy to 24% (52%) by 2035 (2050). Electrification of transport (both light- and heavy-duty vehicles) and industry (e.g., process heat) represent the largest contributors to load growth (Fig. 1, top). Note that the levels of electric vehicle deployment in the High Electrification scenario more closely resemble anticipated electrification based on projected market trends (Supplementary Fig. 13). Electrification is also occurring in buildings, but efficiency improvements lead to roughly offsetting effects in terms of total electricity demand. These levels of electrification are comparable to other deep decarbonization studies in the literature (Supplementary Fig. 14).

The carbon intensity of electricity generation declines over time in all scenarios (Fig. 1, middle). In the absence of national $CO_2$ policies, natural gas and renewables grow, while existing coal and nuclear are gradually retired. Unlike other studies, higher electrification does not lead to increases in dispatch from existing coal in these scenarios even without a decarbonization policy, as the endogenous investment and retirements lead to declining coal generation, which contrasts with dispatch-only short-run marginal emissions studies[20]. Load growth in the High Electrification scenario is largely met with expanded gas and wind generation, which has a significantly lower emissions profile, both in terms of $CO_2$ and criteria pollutants, than the current generation mix. When carbon pricing is added, coal is phased out decades earlier; most natural gas is equipped with carbon capture and storage (CCS) or is co-combusted with hydrogen; nuclear remains in the mix; and solar and wind see much larger increases. Increasing the share of electricity in final energy consumption can facilitate reductions in $CO_2$ and air pollutant emissions economy-wide, even with substantial

electric load growth, particularly when combined with a shift away from combustion-based electricity generation (Supplementary Fig. 18).

CONUS-wide emissions are shown in Supplementary Figs. 3 and 4 and Table 1. Economy-wide $CO_2$ emissions decline across all scenarios (Fig. 1, bottom), though the rate and extent vary by scenario. $CO_2$ reductions relative to 2005 levels are 33% in the Limited Electrification scenario, 44% in the High Electrification, and 78% in the High Electrification with Carbon Price scenario. In April 2021, the U.S. updated its pledge as part of the Paris Agreement to reduce emissions by 50–52% by 2030 from 2005 levels[43]. As shown in Supplementary Fig. 19, the High Electrification with Carbon Price scenario is consistent with this target, while the other scenarios fall short, entailing 19–28% $CO_2$ declines by 2030. $NO_x$ emission reductions are 28% from 2016 to the 2035 Limited Electrification scenario due to on-road and off-road fleet turnover and cleaner fleet mix in the future year. $NO_x$ emission reductions are accelerated in the 2035 High Electrification scenario for on-road and off-road sectors, even with growing population and economic activity, resulting in net $NO_x$ reductions of 58 and 46% with and without carbon pricing, respectively. Emission decreases from the electric sector result mainly from declining emissions from coal generation (Fig. 1, bottom). Under the High Electrification with Carbon Price scenario, electric generating unit (EGU) emissions drop significantly from the 2016 levels for $SO_2$ (99%) and $NO_x$ (82–87%). VOC emission reductions from on-road vehicles and off-road sources are reduced more than 80% in the High Electrification scenarios, but they are offset by emission increases from oil and gas activity and other nonpoint sources, such as solvent utilization and commercial and residential fuel combustion. Total primary PM emissions, which are dominated by fugitive dust emissions from roads, agriculture, and construction, change only by 3–6% from 2016 (Supplementary Table 1). Dust emissions (except for paved road dust) are held constant at 2016 levels with an assumption that future controls will offset growth (consistent with EPA procedures in the 2016 emission inventory). Activity growth in the agriculture sector (livestock waste and fertilizer application) results in $NH_3$ emission increases of about 20% in 2035 and 30% in 2050. Overall, categories of emissions that are dominated by fuel combustion exhibit larger declines with electrification and decarbonization (e.g., $CO_2$, $SO_2$, $NO_x$) than emissions with substantial non-combustion sources (e.g., PM, $NH_3$).

## Air quality results

Air quality model results from CAMx are shown in Fig. 2 for the ozone design value (DV) metric under the NAAQS, i.e., the fourth highest MDA8 ozone concentrations, the form of the standard that with specified exposure levels is deemed protective of primary (human health) and secondary (welfare) effects, currently set at 70 ppb for both endpoints. DV changes in future years were calculated according to U.S. EPA guidance[44] by using model results to adjust the historical DV for the baseline scenario. Deep $NO_x$ emission reductions (ranging from 28 to 67% from 2016) lead to striking ozone DV reductions across the CONUS, though the magnitude of response varies regionally. In particular, the geographic distribution of benefits is highest for states in the Northeast, Southeast, and Ohio River Valley. Ozone benefits due to vehicle fleet turnover in the 2035 Limited Electrification scenario are widespread leading to attainment of the 70 ppb NAAQS in the eastern states but not all of the western states (e.g., California). The western states experience higher background ozone originating from global natural (e.g., wildfires, biogenic, lightning $NO_x$, and stratosphere-troposphere exchange) and international precursor sources[45] making it more difficult for these areas to meet the NAAQS. Our discussion below focuses on regions outside California, as prior studies have addressed California with state-specific assumptions[37,46]. High Electrification without carbon pricing can further reduce ozone by 3–13 ppb in 2035. The impact of carbon pricing in 2035 is most evident in

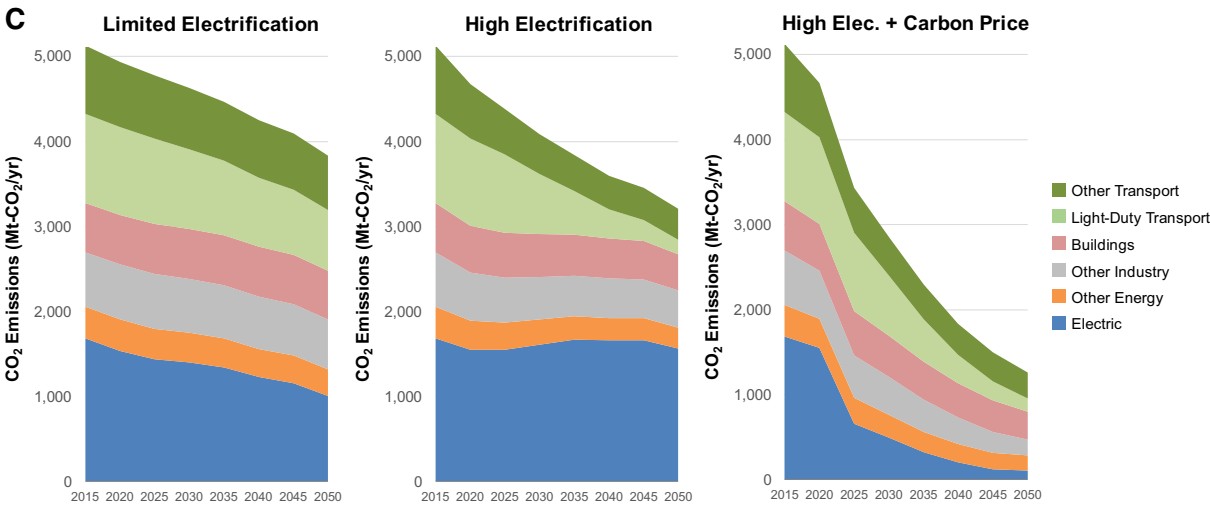

**Fig. 1 | Electricity demand, generation, and economy-wide CO₂ emissions across the three electrification and policy scenarios. A** Electricity demand by end use over time with newly electrified loads at the top. **B** Generation by technology over time (where NGGT are natural gas turbines, NGCC are combined cycle units, and CCS are gas-fired units with carbon capture and storage). **C** CO₂ emissions by sector over time (where Other Transport includes all transport other than light-duty vehicles, including both on-road medium- and heavy-duty trucks as well as the off- or non-road segments). Results are outputs from the US-REGEN model[41].

**Table 1 | Summary of key features and assumptions of US-REGEN**

| Features | Description | Documentation |
|---|---|---|
| **Model resolution and aggregation** | | |
| Temporal resolution | ■ 120 representative hours for power sector investments and operations<br>■ 8760 (hourly) for end-use electricity demand | Section 2.5; Blanford et al.[40]; Bistline et al.[59] |
| Time horizon | 2015 through 2050 in five-year time steps | Section 1.1 |
| Spatial resolution | 17 state-based regions (Supplementary Fig. 1) | Supplementary Note 1 |
| Geographic coverage | Lower 48 U.S. states | Section 1.1 |
| **Model structure** | | |
| Model structure and solution approach | ■ Power sector model is an intertemporal optimization with foresight (i.e., linear program that minimizes the net present value of total system costs of investment and dispatch)<br>■ End-use sectoral models are lagged logit choice simulation models with explicit structural classes and endogenous adoption (passenger transport, buildings, other transport, industrial sub-sectors)<br>■ Retail market model for distributed energy resources uses a simulation approach | ■ Power sector: Sections 1.2 and 2.7; Merrick et al.[60]<br>■ End-use: Section 3; Bistline et al.[59]<br>■ Distributed resources: Section 3.6 |
| Developing organization | Electric Power Research Institute (EPRI) | |
| **Supply-side assumptions** | | |
| Electric sector technologies | Cost and performance projections for generation technologies, transmission, and energy storage are based on EPRI's Integrated Technology Generation Options report | Sections 2.3, 2.4, and Appendix A |
| Refining and upstream energy | Represents industrial activity involved in the conversion of primary fossil energy into refined products for final energy use, including oil refining and natural gas processing and pipeline use, which scales with national consumption | Section 3.5 |
| Negative emissions | Bioenergy with carbon capture and storage (BECCS) in the power sector only | Appendix B; Bistline and Blanford[61] |
| CCS availability | Yes, for fossil fuels and biomass in electric and non-electric sectors; endogenous $CO_2$ transport and storage decisions | Section 2.3.3 |
| Hydrogen availability | Power sector only with endogenous production | Section 2.4.6 |
| **Demand-side assumptions** | | |
| Service demand | ■ Population, economic activity, and service demand are exogenous projections based on U.S. EIA's Annual Energy Outlook<br>■ Service demand price feedbacks from computable general equilibrium model not included in this version | Section 3 |
| End-use technologies | Cost and performance projections for end-use technologies are based on EPRI research | Sections 3.2, 3.3, and 3.4; Bistline et al.[59] |
| Energy efficiency | Exogenous technology-specific improvements; endogenous fuel switching; endogenous price feedbacks (i.e., additional efficiency improvements via increased capital investments in response to higher delivered fuel prices) | Section 3; Bistline et al.[59] |
| Electricity load shapes | ■ For building heating and cooling, based on equipment decisions and hourly temperature profiles<br>■ Passenger transport shapes are constructed based on a combination of exogenous charging patterns that can vary by driver type, day type, and charging location<br>■ For industry and other building uses, shapes are based on EPRI's load shape library: https://loadshape.epri.com/ | Section 3; Bistline et al.[59] |
| Industrial sector disaggregation | Disaggregation based on categories from U.S. EIA data | Section 3.4 (Tables 3–7) |
| **Other assumptions** | | |
| Fuel prices | ■ Endogenous prices for electricity and hydrogen<br>■ Exogenous coal, gas, petroleum, and uranium prices from U.S. EIA Annual Energy Outlook<br>■ Endogenous prices for biomass based on supply curves from FASOM-GHG model | ■ Electricity and hydrogen: Sections 2.2.2 and 2.4.6<br>■ Coal, gas, petroleum: Section 2.4.2<br>■ Biomass: Section 2.4.2 and Appendix B |
| Weather year | 2015; same underlying meteorology and temperatures are used in the end-use model and resource data (e.g., wind and solar output) to develop hourly load shapes to avoid dampening variance through multi-year averaging | Section 2.4 |
| Unit commitment Costs and constraints | Not explicitly modeled for this analysis | Section 2.8 |
| Non-$CO_2$ GHGs and land use | Not explicitly modeled for this analysis | Appendix B |

Documentation lists applicable sections in the full model documentation[41] along with related papers.

the Midwest and eastern Texas, which show additional ozone reductions of 6–10 ppb compared to 2–4 ppb in other areas. This is due to substantial decreases of $NO_x$ emissions from EGUs (75% nationally) and reduced oil and gas activities (19% nationally). Ozone in urban areas declines faster than in rural areas because on-road vehicles are key in driving the overall $NO_x$ emission reductions, and these sources are more concentrated in urban areas. However, total $NO_x$ emissions remain mostly higher in urban areas and peak ozone DV locations remain here in the future.

These scenarios show that electrification leads to air pollution benefits, though these benefits are amplified by carbon pricing policy. The air quality improvements shown in Fig. 2 are substantial even when power sector only moderately decarbonizes. This finding suggests that near-term substitution of fossil fuels for electricity could yield immediate benefits for human health, though the impact varies regionally and by end-use application. Benefits from improved air quality occur rapidly following emissions reductions with a response that is spatially close to where mitigation happens, which differs from

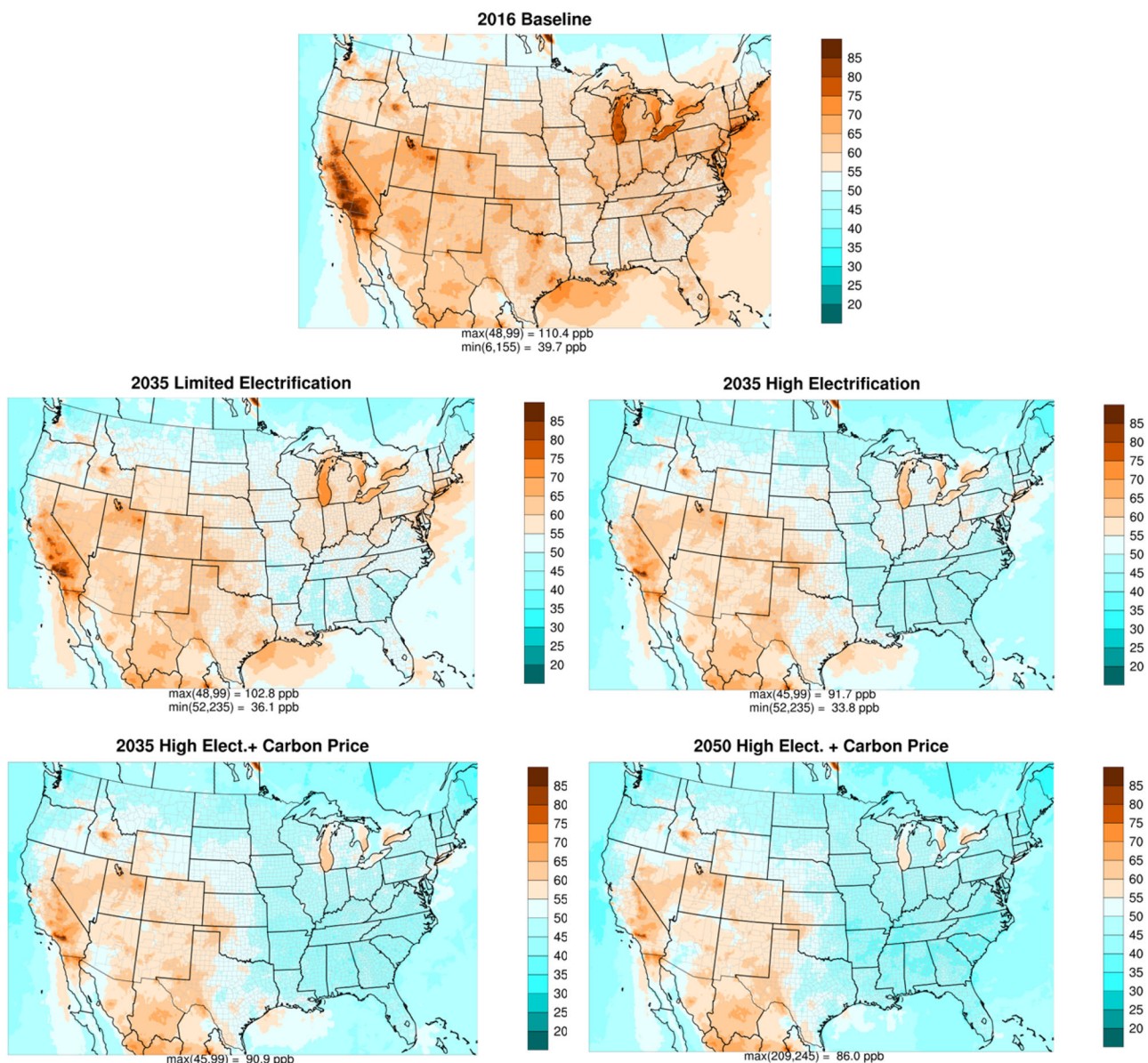

**Fig. 2 | Estimated ozone design values (in parts per billion, ppb) throughout the Continental U.S.** The design value metric under the U.S. National Ambient Air Quality Standards (NAAQS) is the fourth highest maximum daily 8-h average ozone concentrations. A 2016 baseline (top row) is compared with 2035 and 2050 scenarios (bottom rows) with different levels of electrification and policy. Supplementary Fig. 7 shows differences relative to the 2016 baseline.

benefits from decreases in climate damages that may take decades to be felt and may occur in geographically distant locations.

Historical ozone DVs show a declining trend because emissions of ozone precursors continue to decline from 2002 levels, but progress has been slowing in recent years or even reversed in some locations. An example is presented in Fig. 3 for the Aldine monitor in Houston (Texas). The ozone DV at Aldine declined from 100 ppb in 2003 to a low of 72 ppb in 2014 but plateaued around 80 ppb since then, i.e., a lack of progress towards attainment of the 70 ppb NAAQS. Other monitors in Texas and other areas show similar long-term ozone declines that became flat recently (Supplementary Fig. 8). On-the-books strategies lead to the projected 2035 DV of 74 ppb at Aldine and marginally below 70 ppb in West Alton (Missouri), Alsip (Illinois), Dallas (Texas), and San Antonio (Texas), shown in SI. The High Electrification scenario lowers 2035 DV to 67 ppb (7 ppb lower than the Limited scenario). Adoption of carbon pricing lowers ozone by another 4 ppb. The ozone benefit from carbon pricing in 2035 is about

10 ppb in northeast Texas (near Cotton Valley Sand and Haynesville Shale oil and gas activities) compared to 2–4 ppb in other areas. Increased levels of electrification can double or even triple ozone improvement from 2016 as seen in the 2035 and 2050 with carbon pricing scenarios. In 2050, many monitors such as Fulton (Georgia) and Capitol (Louisiana) approach levels that are considered background[45].

Annual $PM_{2.5}$ DV, defined as the annual mean averaged over three years, declined gradually from 2002 but increased more recently at some monitors (Supplementary Figs. 9–12). Current annual $PM_{2.5}$ DVs (2018–2020) at monitors in the eastern U.S. meet the 2012 NAAQS of $12\,\mu g\,m^{-3}$ but many are above $10\,\mu g\,m^{-3}$. Harvard Yards, which is a controlling monitor in Cleveland (Ohio), had a 2017–2019 DV of $11\,\mu g\,m^{-3}$. The 2035 Limited Electrification scenario reduces $PM_{2.5}$ DV from 2016 by $0.3\,\mu g\,m^{-3}$ at this monitor and only up to $0.5\,\mu g\,m^{-3}$ elsewhere (Supplementary Figs. 9–11). $PM_{2.5}$ DV increases from 2016 at some monitors such as at Clinton in Houston (Texas) and Granite City

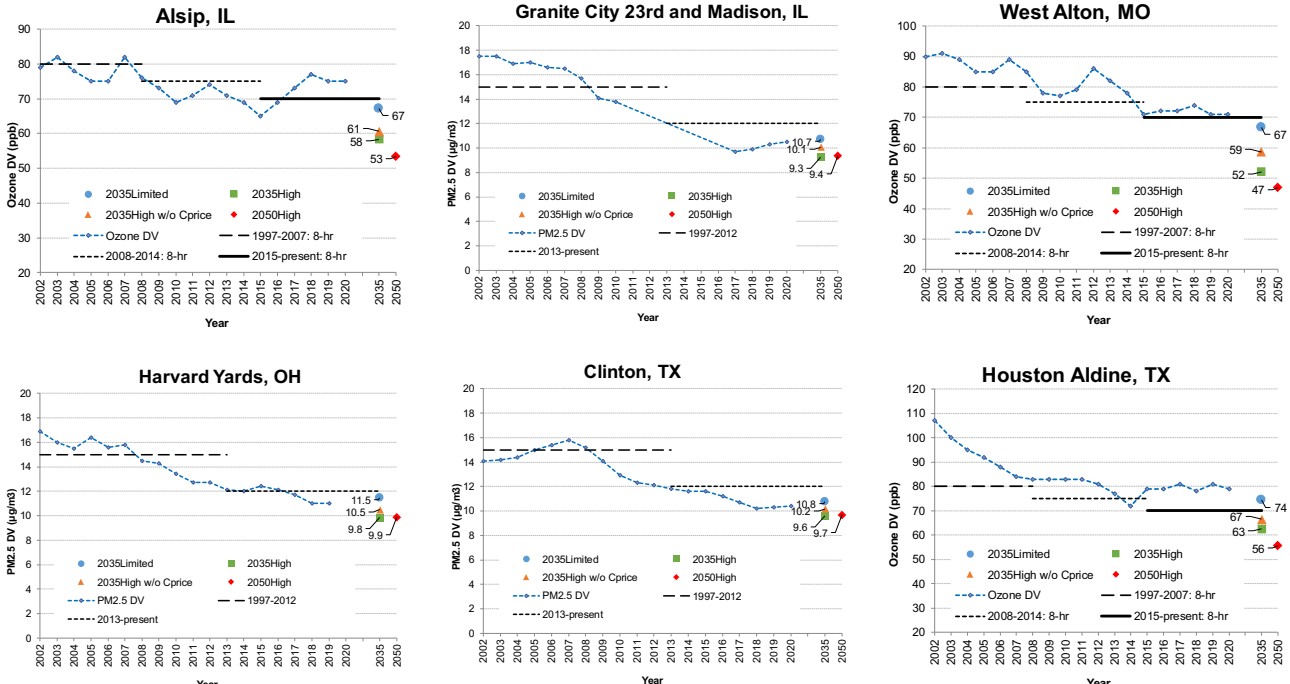

**Fig. 3 | Historical (dotted line) ozone design values (DVs) at Houston Aldine (Texas), West Alton (Missouri), Alsip (Illinois) monitor (top row) and annual PM$_{2.5}$ DVs at Harvard Yards (Ohio), Clinton (Texas), and Granite City (Illinois) monitor (bottom row).** Also shown are the relevant U.S. National Ambient Air Quality Standards (NAAQS) for ozone and PM$_{2.5}$ which changed over time (dashed lines). Recent observed DVs are compared with projected DVs for 2035 and 2050 scenarios (symbols) with different levels of electrification and policy.

(Illinois). PM$_{2.5}$ DV further decreases in the 2035 High Electrification without Carbon Price by 1.0 µg m$^{-3}$ at Harvard Yards, however, most areas see PM$_{2.5}$ DV decreases of <0.5 µg m$^{-3}$. Carbon pricing further lowers PM$_{2.5}$ DV by 0.7 µg m$^{-3}$ at this monitor in line with 0.5–1.0 µg m$^{-3}$ elsewhere. PM$_{2.5}$ DVs in 2050 are similar to 2035 with a carbon price due to increases of primary PM$_{2.5}$ emission from industrial sources (such as metal production and mineral products) offsetting decreases of PM$_{2.5}$ precursor emissions (NO$_x$, SO$_2$).

PM is comprised of many chemical components including both organic and inorganic particles. Decreases in PM$_{2.5}$ DV from 2016 are driven by reductions of both primary (elemental carbon) and secondary PM$_{2.5}$ species (sulfate and ammonium), which decrease in the 2035 Limited Electrification and even more in high electrification scenarios (Supplementary Figs. 9–11). PM$_{2.5}$ nitrate decreases in all high electrification scenarios but increases in the Limited Electrification scenario in wintertime along the Ohio River Valley, where many EGUs are situated and in Texas. Other modeling studies also have found that SO$_2$ emission reductions can lead to PM$_{2.5}$ nitrate increases in the eastern U.S. (e.g.,[47]) although we note that comparisons of decadal trends in modeled and observed PM$_{2.5}$ suggest that models may overstate this effect, possibly because widely used algorithms for modeling the sulfate-nitrate-ammonium aerosol system are biased. If our modeling tends to overstate PM$_{2.5}$ nitrate increases caused by SO$_2$ reduction, then our estimated benefits of electrification will tend to be understated.

Organic aerosol decreases only marginally, and crustal material increases due to increase in the primary PM$_{2.5}$ emissions. While the combustion-related PM$_{2.5}$ emissions from electrified sources decrease, growing population and economic activity are expected to maintain or even increase emissions from multiple source categories such as road dust (tire, brake pad, and roadway treatment), commercial wood/biomass boilers, and industrial facilities (metal production and mineral products). Reductions of emissions from several activities in addition to combustion-related emissions may be needed to further reduce

PM$_{2.5}$. Although PM-related impacts are relatively small on a percentage basis, even small changes in PM$_{2.5}$ concentrations can have a large response in health benefits[19].

## Insights gained from detailed modeling

This analysis finds that electrification decreases CO$_2$ and other air pollutant emissions even in the absence of a national climate policy, and these declines are amplified by decarbonization policies. These findings contrast with earlier studies suggesting more limited emissions effects of electrification, especially those that use short-run marginal emissions estimates[10,20], which characterize marginal emissions from fixed electricity systems and do not account for structural changes over time as many energy systems models do. To illustrate how detailed energy systems modeling facilitates these findings, this section compares our results with simplified approaches using emissions factors that are common in the literature and in actual decision-making[48]:

- REGEN observed: Net emissions effects are based on outputs from a detailed structural model (REGEN) of electric sector capacity planning and dispatch as well as end-use adoption. These correspond to emissions changes in earlier sections. Many systems modeling methods endogenize CO$_2$ emissions from the power sector (and all other sectors) and implicitly use this approach.
- REGEN average emissions: Net effects based on annual average emissions rate outputs from REGEN. These dynamic emissions rates vary over time with changing resource mixes and by model region. Average emissions rates are often used for their simplicity[49], though they omit changes that act on a system's margin, where the generation response to new or existing loads may differ from the current average mix[48].
- Constant average emissions: Net effects use average emissions rates, which are assumed to remain constant over time. Annual 2019 values come from ref. 20 at the interconnect level.

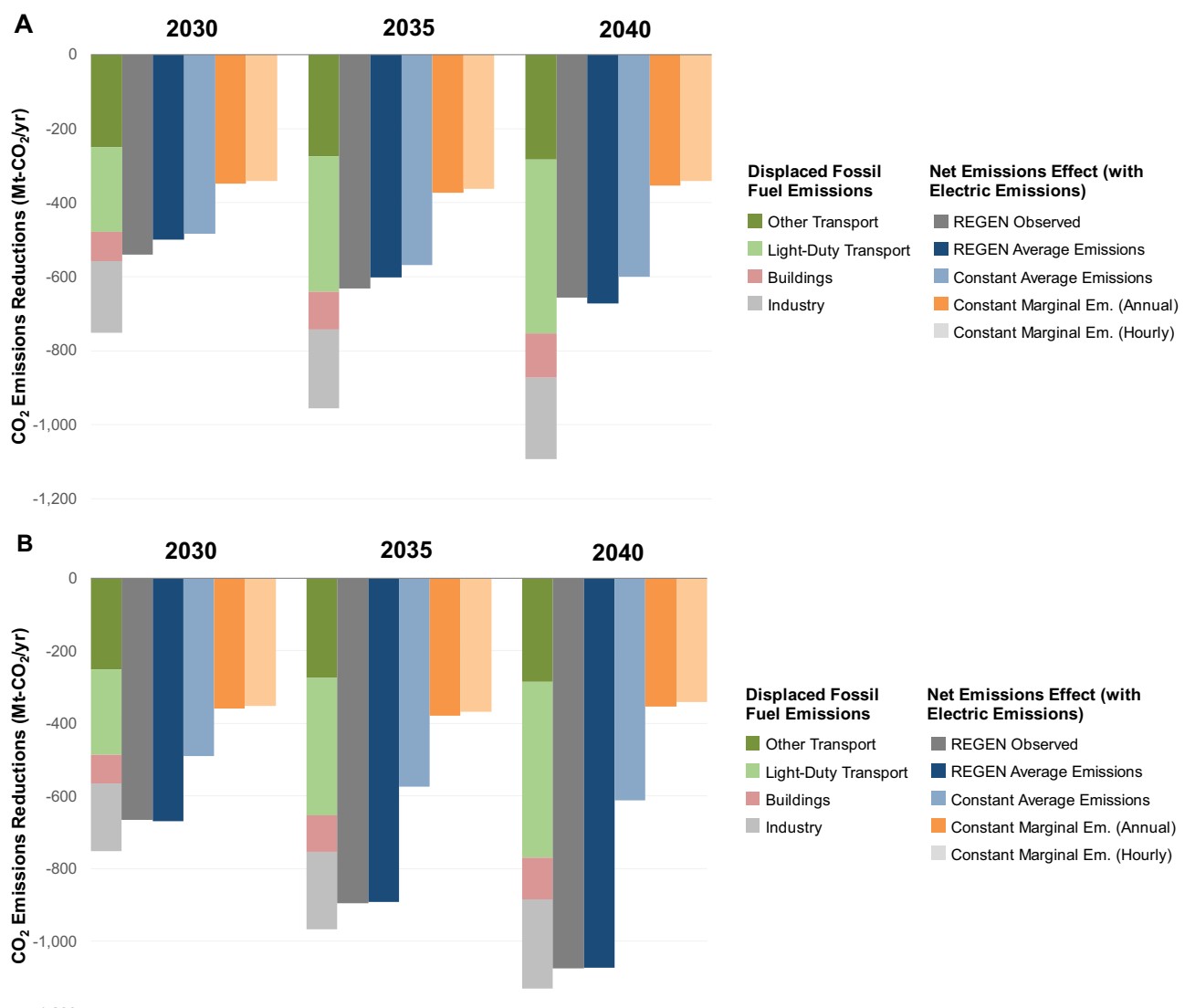

**Fig. 4 | CO$_2$ emissions reductions from electrification over time across different approaches for accounting for electric sector emissions.** Scenarios assume high electrification with current policies (**A**) and economy-wide carbon tax (**B**). Stacked bars show displaced fossil fuel emissions by sector. The other bars show the net emissions effect after accounting for emissions associated with increased electricity production under five different assumptions. Constant average and marginal emissions are based on 2019 estimates[20].

- Constant marginal emissions rates (annual): Net effects use short-run marginal emissions rates from ref. 20, which are assumed to remain constant over time. Annual 2019 values are used at an interconnect level. Short-run marginal emissions rates estimate how new loads would be served from the current grid but do not account for how new loads could induce investments or retirements of assets[48].
- Constant marginal emissions rates (hourly): Net effects use 2019 marginal emissions rates from ref. 20, which are reported at hourly levels.

Figure 4 shows CO$_2$ emissions reductions from higher electrification across these different methods. Marginal and average emissions approaches systematically underestimate reductions from electrification: These emissions factors only capture 52 to 91% of anticipated CO$_2$ reductions under a reference scenario and 32 to 74% of anticipated CO$_2$ reductions under a carbon pricing scenario. Marginal emissions estimates implicitly assume that coal and gas generation are on the margin for large fraction of hours[48] and do not capture changes over time. Average emissions metrics can better approximate anticipated emissions changes, especially dynamic ones based on model outputs that account for changes in the grid mix over time.

The emissions bias from simplified emissions factor methods is larger under scenarios where structural change is expected in the electric sector, especially with stringent CO$_2$ policies. Structural modeling methods, such as the approach in this paper, are particularly important in settings where system changes are expected over the life of the asset. Constant average emissions perform better than marginal emissions but worse than observed reductions for a carbon tax, since the electricity generation fleet decarbonizes over time. Supplementary Fig. 20 illustrates the rapid decline in emission intensity of electricity generation over time in the carbon price scenario.

Overall, these comparisons indicate that emissions declines from electrification are larger than simplified marginal and average emissions methods suggest. Emissions decline with today's grid mix but are lower with future market trends (e.g., coal retirements, renewables) and existing state policies. End-use electrification can be environmentally beneficial today and increase over the lifetime of the device.

## Discussion

Our results show that electrification can substantially and broadly improve air quality in the U.S. and is an effective attainment strategy. The air quality benefits of electrification and decarbonization policy are immediate and sizable, especially given improved understanding of human health impacts of air pollution exposure[50]. The quick air pollution response by 2035 with localized benefits contrasts with the comparatively slow response and global benefits due to climate change mitigation, where longer timescales for atmospheric and ocean warming mean that the largest reductions in climate damages occur after 2050[6,51]. Electrification can double or even triple the ozone improvement relative to 2016 as seen in the 2035 and 2050 with carbon pricing scenarios. Deep $NO_x$ emission reduction (42 and 54% reduction from the Limited scenario) is key to achieving such ozone benefits. On-the-books strategies relying on fleet turnover to cleaner conventional vehicles and equipment may not be sufficient to meet the NAAQS particularly if standards continue to tighten.

Lowering $PM_{2.5}$ presents a different challenge because $PM_{2.5}$ is made up of several components, not all of which are related to fuel combustion. We find that the complex interaction between particulate nitrate and sulfate weakens the $PM_{2.5}$ benefit of $NO_x$ emission reductions in the Limited Electrification scenario. In the High Electrification scenario, $NO_x$ and $SO_2$ emissions fall further, minimizing this effect so that secondary $PM_{2.5}$ formation decreases. Meanwhile, growing activity from non-combustion sources could partially offset the electrification benefits by increasing primary $PM_{2.5}$ emissions.

Our modeling framework helps derive these important NAAQS-relevant air quality outcomes. The technological detail and endogeneity of technology adoption in the US-REGEN model assure a robust and internally consistent representation of the evolution of emissions drivers in our future energy scenarios. For example, the linking of these two state-of-the-art tools makes it possible to quantify air quality impacts at a sub-sector level for the transportation and industrial sectors, taking into account simultaneous changes in the electric power sector. The full-form AQM is necessary to address nonlinear chemistry of ozone and PM seen in this work due to drastically changing emission profiles, spatially and temporally, for distinct chemical species due to electrification and decarbonization. Simpler approaches, such as LCA or reduced-form modeling (which often do not account for structural and technological changes in the linked sectors and activities), may be unable to represent the response of the atmosphere to such changes, in particular when estimating future design values using established methods. While earlier energy systems decarbonization modeling is consistent with these results and the air quality insights also are directionally consistent, the linked nature of our modeling is a unique contribution of our study, which provides a more complete portrait of these linked transformations.

While the results of this study demonstrate the significant co-benefits of end-use electrification and electric sector decarbonization for local and regional air quality, and thus human and ecosystem health, some outstanding questions remain. First, it is quite possible that U.S.-regulated air pollutant emission standards will be further tightened between now and 2050. In that case, emissions reductions from electrification may not be enough to drive air pollutant concentrations down to the newly required levels everywhere. That would imply additional measures being needed for non-electrified end-uses and non-combustion sources: e.g., improvements in combustion-based emission control technologies (whether fossil or biomass); development of lower-volatility, less reactive solvents; best management practices for agricultural emissions controls; and advancements in reducing primary $PM_{2.5}$ emissions from industry and fugitive dust from roads and other non-combustion sources. Policies and incentives to achieve these goals could include strengthening regulatory approaches at the federal level under existing Clean Air Act authorities (e.g., EPA's Cross-State Air Pollution Rule to assist downwind states to attain fine particle or ozone NAAQS), tax credits for lower-emitting technologies, and state-level ambient air quality standards. Alternatively, it could suggest that further electrification and accelerated timeframes will be needed to achieve more stringent air quality standards over the long term. Whether electrification would be a more cost-effective alternative at the margin than end-of-pipe or other emissions controls in a future scenario is an open question and would require further study. The bottom panel of Fig. 1 illustrates how residual emissions are highest for sectors with higher marginal abatement costs, including non-light-duty transport (e.g., aviation, shipping), high-temperature heat for industry, and sectoral options on the steep portion of sectoral abatement cost curves (e.g., space heating for buildings in the coldest climates), which aligns with the broader decarbonization literature[52].

Second, while this study assumes a quickly growing economy-wide carbon price in several of its scenarios, it does not explicitly incorporate some of the latest U.S. federal policy proposals for achieving net-zero electric sector by 2035 and economy-wide $CO_2$ emissions in the years post-2035 (e.g., clean electricity standards, vehicle and building appliance mandates, updates to the official social cost of carbon estimates used for rulemaking, etc.). However, reaching net-zero in the electric sector and across the economy may be achieved using a range of approaches that have very different implications for air quality[53].

Third, the results in this paper examined the regional heterogeneity of air quality impacts from electrification. Future work should quantify air quality improvements at the community level to examine environmental justice and should model environmental justice policies, which are central issues for policymakers.

Finally, while our study did not attempt to translate changes in air quality to direct impacts to human health, this is an important next step for quantifying the implications of electrification and decarbonization. The approach employed in the current study—running US-REGEN in combination with CAMx—could be used in future work to address these and other outstanding questions.

## Methods

### Energy system model

To examine the roles of electrification and decarbonization on emissions, this analysis uses the 2020 version of EPRI's U.S. Economy, Greenhouse Gas, and Energy (US-REGEN) model, which features an electric sector capacity planning and dispatch model linked to an end-use model with technological, temporal, and spatial detail. Regional aggregation is shown in Supplementary Fig. 1. REGEN is fully documented in EPRI (2020)[41], so only summaries of key features and assumptions are provided here.

Table 1 highlights several key features and assumptions of the model and includes references with additional detail.

The electricity sector model minimizes the net present value of total system costs subject to policy, technical, and market constraints under different assumptions. This model includes endogenous capacity planning and dispatch with joint investment decisions for generation options; energy storage technologies; transmission; hydrogen production; as well as $CO_2$ removal, storage, and pipelines. US-REGEN was built to represent the unique economic and operational characteristics of variable renewables, energy storage, and other technologies as well as the policies that support them. Cost and performance estimates for different supply-side technologies come from the literature, expert elicitations, and EPRI's Technology Assessment Guide, which are provided in the US-REGEN documentation available at: https://us-regen-docs.epri.com.

The end-use model includes structural detail across several dimensions relevant for fuel and technology choice, such as building size, type, and vintage, climate zone and location, and vehicle ownership and driving intensity. Within each structural category, service

demand may be met with a range of options, characterized as combinations of fuels and technologies. The model evaluates the total cost of each option in each new vintage based on assumed technology cost and performance, fuel prices, structural attributes of service demand, and non-economic factors. The resulting allocation across the options is based on a logit model translating relative costs to equilibrium market shares, with a lagged process to simulate a gradual transition toward the model's calculated equilibrium shares. Parameter selection for the logit model is discussed in Supplementary Note 1. The parameterization of the logit model determines the sensitivity of the market allocation (within a given nest in the nested structure) to assumed ("observed") costs, or, conversely the influence of non-modeled ("non-observed") or costs or preferences, which by construction are assumed to be independently and identically distributed for each technology in the nest. The model then determines annual and hourly fuel use by region as a function of the resulting mix of end-use technologies. Investment costs for end-use technologies (e.g., light-duty electric vehicles, air-source heat pumps) are sourced from the literature and EPRI expert elicitations. Behavioral features of energy consumers are represented by incorporating disutility costs associated with purchase decisions such as refueling station availability for passenger vehicles[54], using a simulation-based approach to capture unobserved preference heterogeneity across households (Table 1), and explicitly breaking out structural classes to capture observed heterogeneity across households and firms. Note that this framework does not capture possible interactions of heterogeneous agents, where adoption and utilization decisions are influenced by the choices of others, as many agent-based models do[55].

Fuel costs are sourced from the U.S. EIA's Annual Energy Outlook reference case, which also serve as inputs to the electric model. The cost of electricity is an input provided by the electric sector model. The model uses exogenous assumptions from the U.S. EIA's Annual Energy Outlook for projected economic activity, population, and sector-specific service demands (e.g., vehicle miles traveled) over time, which account for current economic, cultural, and technological trends (e.g., gross domestic product roughly doubles between 2015 and 2050, while population increases by 25%). These assumptions are held fixed across scenarios. Data from the U.S. EIA's Annual Energy Outlook can be accessed through their interactive table browser: https://www.eia.gov/outlooks/aeo/data/browser/.

US-REGEN has been applied in dozens of peer-reviewed article and reports: https://esca.epri.com/research.html. An interactive version of the documentation with detailed assumptions and data can be found at: https://us-regen-docs.epri.com.

### Air quality model
We use the Comprehensive Air Quality Model with Extensions (CAMx), version 7.0, for air quality modeling with a 12-km grid covering the entire lower 48 states and nested within a larger 36-km grid (Supplementary Fig. S4) for every hour of calendar year 2016. The meteorology for 2016 is used for all scenarios so that the projected air quality changes can be attributed solely to emissions changes. Our 2016 baseline CAMx simulation was based on the EPA's 2016 version 1 modeling database which EPA used for the Revised Cross-State Air Pollution Rule Update for the 2008 ozone NAAQS. Note that, under higher temperatures in a changing climate, vehicle exhaust emissions and biogenic (e.g., isoprene) emissions may increase, which could be explored in future work.

For 2016, model performance of the maximum daily 8-h average (MDA8) ozone shows normalized error achieving the recommended performance goal of within 15%[56] in all seasons. Normalized bias for MDA8 ozone meets the bias goal of ±5% in summer and fall and achieves the less stringent bias criteria of ±15% in spring and winter. $PM_{2.5}$ performance is variable across seasons and is generally comparable to EPA's 2016 modeling on which our model inputs were

based. Normalized bias achieves the ±30% bias performance criteria in all seasons with a tendency to overestimate, especially in winter (Supplementary Notes 2 and 3). A description of the linkages between US-REGEN model results and the CAMx air quality model is described in the Supplemental Information.

### Scenarios
All scenarios include representations of significant on-the-books federal and state policies and incentives as of June 2021, including:
- State-level renewable portfolio standards, including technology-specific carveouts for solar
- State-level clean electricity standards with state-specific definitions of qualifying resources
- State-level offshore wind mandates (CT, MA, MD, ME, NJ, NY, RI, and VA) and energy storage mandates (CA, NJ, NY, and VA)
- California AB32, represented as a carbon tax based on projections by the California Air Resources Board
- Regional Greenhouse Gas Initiative (RGGI) cap-and-trade system
- Current Clean Air Act Section 111(b) new source performance standards
- Investment tax credits and production tax credits.

The carbon tax scenario layers this economy-wide tax on top of these existing policies. Note that scenarios do not explicitly include the updated U.S. Nationally Determined Contribution pledge to reduce emissions by 50–52% by 2030 from 2005 levels, since formal policies are not yet in place to achieve this target, though the High Electrification with Carbon Price scenario is consistent with this target (Supplementary Fig. 19). Scenarios also incorporate announced retirement dates for coal plants.

The Limited Electrification scenario is a hypothetical benchmark scenario that can be used to quantify the impact of electrification on outcomes of interest. This scenario is intentionally less optimistic about the speed at which consumers adopt electric end-use technologies across transport, buildings, and industry. This is achieved in the modeling by: (1) Exogenously assuming low EV adoption rates in all future years; (2) Imposing a constraint on the electrification of buildings so that shares do not increase beyond today's levels; and (3) Setting higher costs for electric technologies in industry and heavy-duty transport. These assumptions lead to electricity demand remaining roughly flat over time (Fig. 1).

The High Electrification scenarios use more optimistic assumptions about advanced end-use technologies. Higher electrification in this scenario is driven by: (i) allowing EVs to deploy endogenously (given continuing trends of falling battery costs); (ii) accelerating the performance assumptions for heat pumps; and (iii) faster turnover rates for the existing equipment stock. The High Electrification with Carbon Price scenario adds an economy-wide carbon prices across all model regions that starts in 2025 at $50/tCO_2$ (in 2020 USD), which grows at 7% per year (reaching $271/tCO_2$ in 2050). This price is intended as a proxy for a suite of $CO_2$ policies for the electric and end-use sectors. The starting price in 2025 is comparable to the proposal by the Climate Leadership Council in 2019 (which was proposed to start at $40/tCO_2$ in 2017 USD by 2021 and then increase annually at 5% above inflation[57]) and multi-model scenarios for the U.S.[43,58], including a recent study of the 2030 U.S. climate target that estimates a marginal cost of $CO_2$ reductions between $36 to $155/t-CO_2$ by 2030[43].

All energy system scenarios in US-REGEN are run in 5-year intervals through 2050. However, due to the computational and resource intensity of runs, air quality scenarios in CAMx are only run for select years and scenarios of interest. The analysis focuses on 2035 to explore the impact of electrification over a time horizon that allows for significant penetration of technologies while at the same time being informative to the deadlines of upcoming State Implementation Plans, which would be necessary for attainment demonstration of potential revisions to the NAAQS for ozone and $PM_{2.5}$.

Detailed scenario assumptions are provided in Supplementary Note 6.

## Data availability

Source data underlying all figures and all other non-proprietary data supporting this study are available from the corresponding author upon reasonable request.

## Code availability

CAMx source code, version-specific inputs, and user guides are available at https://www.camx.com/download/. US-REGEN model documentation, input assumptions, and links to papers with model source code are available at https://esca.epri.com/models.html.

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

## Acknowledgements
The views and opinions expressed in this paper are those of the authors alone and do not necessarily state or reflect those of their respective institutions.

## Author contributions
J.B., G.B., J.G., E.K., D.M., U.N., H.S., T.S., and G.Y. designed the research and methodology; G.B., J.G., D.M., U.N., H.S., and T.S. performed the research; E.K., J.G., D.M., U.N., H.S., T.S., and G.Y. analyzed the data; J.B., G.B., J.G., E.K., D.M., U.N., H.S., T.S., and G.Y. wrote and edited the paper.

## Competing interests
The authors declare no competing interests.
