## [Peer review file · Nature Communications]

REVIEWER COMMENTS

Reviewer #1 (Remarks to the Author):

Bistline et al.'s manuscript, "A Full Economy-Wide Evaluation of CO₂ and Air Quality Impacts of Electrification in the United States" presents an extremely examination of future electrification on CO₂ emissions and air quality using two state-of-the-art systems, US-REGEN and WRFCAMx. They have built on previous work that tends to neglect more nuanced features of the energy-emission interplay that can only be captured with a full energy dispatch model. Moreover, their full year simulations at relatively high resolution using a full air quality model represents a one of the best attempts to characterize air quality impacts from future electrification. While I have a few questions and suggestions, the paper is extremely well written and I recommend publishing following only minor revisions.

Figure 2: Add difference plots

L41-42: Could mention connection to ozone and PM.

L46: "fugitive dust" is mentioned several times, can you explain the difference/relation to 'natural', windblown dust?

L72, 78-79: Some previous studies ran sensitivities with different adoption, market shares, and /or generation types, no?

L210: remove "the" preceding outside California

Does the REGEN model take into account population change and/or changes in vehicle miles traveled? (e.g., more remote work). What are population assumptions?

I think more details could be provided on a seasonal analysis; e.g., some places experience peak PM_{2.5} in winter, elsewhere in summer, and so an annual average – while important for health metrics – misses out on some key

You have Figure S8 showing DV for PM2.5 over the Midwest, but why not having a full CONUS figure? Additionally, the DV for ozone is a high percentile value, while the DV for PM2.5 is a much more 'smoothed' statistic, could you show something similar for PM2.5 (e.g., the 95th or 99th percentile)?

It would be helpful to add a brief discussion of how climate change (mainly increasing temperatures) would impact emissions (e.g., biogenics, exhaust)

I appreciate the full Table of model performance metrics, and although all criteria are within acceptable limits, I'm curious how the model performs by region. One suggestion would be to include ozone and PM2.5 validation at the specific sites/monitors mentioned in the text,

I think the main text could use a bit more discussion of VOC emissions changes

Since it always seems to be a caveat picked up on, how are manufacturing (battery) emissions included?

Can you please provide more details on the WRF simulation performance and how that may have impacted your simulations? (the link provided in the reference is not functional)

Given the offline nature of WRF-CAMx, a brief discussion of the uncertainty in including aerosol feedbacks might be helpful.

Given the proprietary nature of REGEN, it will be difficult to repeat this type of analysis, will the emissions be available?

Given the population density and fleet of metro regions, a few more statistics on these regions might be warranted.

Reviewer #2 (Remarks to the Author):

Overview:

This paper proposes a combined modelling approach to understanding the impact that energy system electrification can have on AQ in contiguous United States. The authors combine an energy system model, including a dispatch and capacity planning model linked to a demand model, with a full-form photochemical air quality model of the contiguous United States.

The paper is well written, relatively clear, and well structured. The approach is designed to transfer emissions trajectories from the energy model results to a full AQ model to allow the modellers to say something about the local concentrations of criteria pollutants and understand how different scenarios improve or worsen air quality. This has an important future link to human health (not addressed) which is mentioned, and represents an approach that I have not seen before. It also responds to the issue that regional energy models are often unable to say anything about the impact of their emissions on local AQ because of the lack of granularity and the very local nature of AQ impacts of emissions. Setting up such a methodology - or the first steps of it - is important work.

I will caveat my statements by the fact that my modelling expertise is restricted to energy system modelling. As such my comments will not touch on the AQ modelling approach or assess their suitability.

Noteworthy results:

Results suggest the following:

- That electrification and decarbonisation policies have the potential to significantly improve AQ.
- That this improvements has limits linked to what the source of the AQ emissions are and that tighter carbon constraints for example may not lead to further improvements in AQ beyond a certain plateau.
- That methods that apply constant emissions estimates will tend to under-estimate the emissions reduction potential under different decarbonisation scenarios.
- That emissions reductions and AQ improvements are possible without decarbonisation policy - i.e. under optimistic assumptions about advanced end-use technologies.

These results are sound and make sense in line with what other energy system modelling results would suggest. The dynamics that the scenarios highlight here are not surprising or novel from an energy system modelling perspective. Equally, while not an expert, I feel that the operation of the AQ model in itself yields expected dynamics in relation to pollutant circulation, secondary pollutant formation, etc. I believe that novelty in this work lies in the linking of the models and the ability to complete the picture of the energy model results with a detailed translation of their local impact. I would therefore re-focus part of the discussion to make this much plainer to the reader.

Similarly, on the notion that decarbonisation policies can support AQ emissions but that this will plateau - and results from the AQ modelling showing that different parts of the US will face different persistent problems that may not meet NAAQS. Could the authors consider making recommendations or

comments on the energy policies that could, if implemented in chosen areas, be shown to tackle this problem? (I realise this is partially alluded to with the paragraph p13 line 380 - but this only states what is missing in terms of existing policy rather than venturing a suggestion as to what types of policies in what areas and over what time-frames could be tested to remedy these tougher AQ emission issues.)

Energy Modelling

Having reviewed the US-REGEN model documentation (briefly!) and considered the paper and the SI - I'm afraid I feel that the information provided on the energy model and the scenarios that are presented here is insufficient.

Scenarios:

- 1) While the description of the scenarios is clear enough to give the reader an understanding of how they differ, their key assumptions are not listed. This makes it difficult to i) understand in absolute terms what "more optimistic assumptions" or "accelerating the cost and performance assumptions" means and whether this is realistic. I would invite the authors to provide additional SI information on the key assumptions that differentiate the scenarios one from the other.
- 2) Carbon price data is explicit for the third case, but is not "justified" or put into context: 271\$ is a very specific number, where does this come from and why? Also, how does this compare to future cost of carbon projections in other US scenario exercises?
- 3) What constraints climate policies - if any - do these scenarios all include? This relates to my question about e.g. the US NDC which appears lower down.

Energy model:

- 1) Understanding what core data underpins / is found in all scenarios and is important for the results is not easy. Reviewing the model documentation (which is extensive - and beyond much of the documentation that is available for other modelling frameworks!) there are tables of costs and efficiencies, yes, but there are also different sets of assumptions for different technology options - and the potential to include technologies or not. Along with the SI information for the scenario points it would therefore be welcome to have, at least, a clear statement of what is included in the model with direct reference to relevant parts of the documentation - both in terms of technology options (including e.g. CCS and DAC for example) and in terms of data used. I understand this is a large undertaking and would consider the authors' selection of what constitutes critical information acceptable.
- 2) One example of this is - what temporal resolution is US-REGEN run on for this work?

Methodology

- 1) My main comment on the methodology pertains to the description and language used to describe the energy model. This language and description has a strong tendency to describe it as "detailed structural energy systems model", "integrated electric and energy system model", "detailed energy systems

model", "uniquely capable" - the list goes on and leads the reader to assume it is a whole energy system model. US-REGEN is a capacity planning and dispatch model of the ELECTRICITY system, combined with a logit model of demand side service delivery. The entire upstream and process sector producing any other commodity but electricity (and H2 I believe), is missing. This has an important impact on the interpretation of the results - particularly since a Logit model will be parameterised against existing macro-economic relationships. This to me creates two issues that are not highlighted clearly, first that a large part of the whole energy system is not endogenised, and second that the demand side model in US-REGEN will to an extent apply our current expectations of the energy system to determine what should be installed or not.

I completely recognise that different modelling approaches exist, and I believe this one has value. However its characteristics need to be made much clearer throughout the paper and the SI.

2) Extending this - please comment on how the logit model is parameterised for demand side technologies that are not yet available or do not yet have extensive data available with a potential reference to the section of the US-REGEN documentation that details this.

3) The baseline scenario is chosen is hypothetical (admittedly, as all baselines and scenarios are), but seems to depart from "current trends". It is not clear in the paper how this reference scenario compares to e.g. current roll-out trends for Evs which here are "limited", or, "imposing a constraint on the electrification of buildings so that shares do not increase beyond today's levels". Could you please provide a paragraph / explanation as to how this future compares to what is currently happening in the US energy system? And an explanation where there is a cutoff (e.g. the case of the buildings) as to why this is appropriate for the modelling exercise?

4) It is not clear why the results from the modelling presented for the different scenarios do not all include 2050. Could you please consider including additional scenario result outputs for the energy system model for 2050 for the limited electrification and high electrification scenarios - or alternately justifying why the choice has been made to present results to 2050 only for the carbon price scenario.

"Insights gained from detailed modelling"

This section discusses the difference in emissions reductions expected if analysis either endogenises electricity emission factors or instead uses an approach that applies fixed emission factors (with different fixing methods). This is interesting and useful. It is however important to note that IAM or whole energy system modelling approaches typically endogenise carbon content of electricity calculations as standard. This therefore - in and of itself - does not represent a step up as compared to other comparable energy system modelling approaches. This does not remove the fact that any analysis that uses fixed emission factors for this will underestimate emissions savings as the electricity sector decarbonises - but is rather a statement that this is an expected result. This section may be more impactful if it stated this more clearly and focused more on the analytical approaches that do use fixed emission factors and are typically used for policy.

Useful additional information

- Consider including data for US emissions breakdown and their trends, and energy consumption by sector and fuel in the SI for reference and comparison
- Please include commentary as to how the scenarios you run compare to / meet current US energy and emissions targets. I understand that different states will have different levels of ambition and that this could be difficult to present, but an understanding of how the scenarios measure up to e.g. US NDC ambitions would help to place the results in context.

Specific comments

- P2 line 33 - other methods for decarbonising energy systems include lowering demand - see (Grubler et al, 2018)
- P2 line 68 "annual snapshots" is not appropriate. While IAMs do have a coarser temporal resolution, they will typically represent seasons X day types X day parts in some detail. They do not present a "single year approach" please amend or clarify.
- P2 line 42 ref [6] it is notable that this does not include eco-system services, the valuation of which is starting to shift this picture and would be worth recognising.
- P3 line 76 "range of scenario specific drivers" please detail where appropriate.
- P3 line 85 "established methods for assessing future AQ" please reference
- P3 line 105 "behavioral" please qualify - this is based on an economic understanding of behaviour rather than e.g. on the parameterisation of an agent based model with different actors represented using observational data.
- P3 line 109: "rather than relying on a historical snapshot" - please include a reference clearly to studies that do this and clarify in text. Dynamic marginal emissions related to electricity production are typically standard in most energy system modelling exercises.
- P4 line 157 "Unlike [...]scenarios" could you please explain why this is the case? What is it about these runs where - as I understand it - no carbon constraints are implemented that is different from these other exercises where coal is maintained/increased?
- P13 line 367 paragraph: could the authors consider discussing the residual non-electricity fuel consumption in different sectors to highlight areas where this "residual emissions" problem is expected?
- P14 line 407 reference [50] search engine results for this send the reader to publications from 2012 - consider adding a link in the reference as I assume this should point to the us-regen-docs.epri.com reference lower down on line 423.

References

Grubler, A., Wilson, C., Bento, N., Boza-Kiss, B., Krey, V., McCollum, D.L., Rao, N.D., Riahi, K., Rogelj, J., De Stercke, S., Cullen, J., Frank, S., Fricko, O., Guo, F., Gidden, M., Havlík, P., Huppmann, D., Kiesewetter, G., Rafaj, P., Schoepp, W. and Valin, H. 2018. A low energy demand scenario for meeting the 1.5 °C target and sustainable development goals without negative emission technologies. *Nature Energy*. 3(6), pp.515–527

Response to Reviewers

Reviewer #1

Comment 1.1: “Bistline et al.’s manuscript, “A Full Economy-Wide Evaluation of CO₂ and Air Quality Impacts of Electrification in the United States” presents an extremely examination of future electrification on CO₂ emissions and air quality using two state-of-the-art systems, US-REGEN and WRF-CAMx. They have built on previous work that tends to neglect more nuanced features of the energy-emission interplay that can only be captured with a full energy dispatch model. Moreover, their full year simulations at relatively high resolution using a full air quality model represents a one of the best attempts to characterize air quality impacts from future electrification. While I have a few questions and suggestions, the paper is extremely well written and I recommend publishing following only minor revisions.”

Response 1.1: We thank the reviewer their assessment of the importance of the issues addressed in our paper and of the contributions of our work.

Comment 1.2: “Figure 2: Add difference plots.”

Response 1.2: We have added a figure in Supplementary Note 4 with difference plots based on this suggestion. We also added a sentence to the caption of Figure 2: “SI Fig. S7 shows differences relative to the 2016 baseline.”

Comment 1.3: “L41-42: Could mention connection to ozone and PM.”

Response 1.3: After the statement

“Currently, the transportation and electricity generation sectors are leading emitters of U.S. CO₂ and also criteria pollutants, including oxides of nitrogen (NO_x) and oxides of sulfur (SO_x).”

We have added

“Their emissions contribute to the formation of common and widespread air pollutants, such as ozone and fine particulate matter (PM). PM can be directly emitted from these sources (primary) or formed from gaseous precursors (secondary).”

Comment 1.4: “L46: “fugitive dust” is mentioned several times, can you explain the difference/relation to ‘natural’, windblown dust?”

Response 1.4: After the statement

“However, these impacts are uncertain due to unknowns about the speed and scale of electrification, the complexity of interactions between emissions and air quality, as well as the fact that other sources of emissions such as fugitive dust, agriculture, and solvent use also affect air quality.”

We have added the following clarification

“(fugitive dust includes emissions from anthropogenic disturbances, such as resuspension of road dust, agricultural dust, and construction dust, whereas natural dust is from wind disturbances on land surfaces)”

Comment 1.5: “L72, 78-79: Some previous studies ran sensitivities with different adoption, market shares, and/or generation types, no?”

Response 1.5: We expanded this passage to include these sectoral sensitivities: “These features provide endogenous responsiveness of electricity demand and hourly electric load shapes to a range of scenario-specific drivers, which improves upon previous efforts with exogenous emissions scenarios, end-use representations, and sectoral adoption sensitivities [32, 37, 6, 38].”

Comment 1.6: “L210: remove “the” preceding outside California”

Response 1.6: We changed this sentence to: “Our discussion below focuses on regions outside California, as prior studies have addressed California with state-specific assumptions [37, 44].”

Comment 1.7: “Does the REGEN model take into account population change and/or changes in vehicle miles traveled? (e.g., more remote work). What are population assumptions?”

Response 1.7: We added sentences to the Methods section on the sources of these parameters: “The model uses exogenous assumptions from the U.S. EIA’s *Annual Energy Outlook* for projected economic activity, population, and sector-specific service demands (e.g., vehicle miles traveled) over time, which account for current economic, cultural, and technological trends (e.g., gross domestic product roughly doubles between 2015 and 2050, while population increases by 25%). These assumptions are held fixed across scenarios. Data from the U.S. EIA’s *Annual Energy Outlook* can be accessed through their interactive table browser: <https://www.eia.gov/outlooks/aeo/data/browser/>.”

Comment 1.8: “I think more details could be provided on a seasonal analysis; e.g., some places experience peak PM_{2.5} in winter, elsewhere in summer, and so an annual average – while important for health metrics – misses out on some key”

Response 1.8: Though seasonal exposure to PM_{2.5} can differ from region to region, we have opted to focus our analysis, as the reviewer notes, on the annual metric which is based on chronic health impacts from exposure to PM_{2.5}. In Table S4 of the Supplemental Information, the model performance metrics are shown to establish model performance at seasonal timescales and separated out by different primarily urban (CSN) and primarily rural (IMPROVE/CASTNET) ambient measurement networks.

Comment 1.9: “You have Figure S8 showing DV for PM_{2.5} over the Midwest, but why not having a full CONUS figure? Additionally, the DV for ozone is a high percentile value, while the DV for PM_{2.5} is a much more ‘smoothed’ statistic, could you show something similar for PM_{2.5} (e.g., the 95th or 99th percentile)?”

Response 1.9: As generating design-value consistent maps for the entire U.S. is a cumbersome task, we have opted to show two additional figures with the PM_{2.5} design values in the regions of key analysis. These are now included in the Supplemental Information (SI Fig. S10 and 11).

Comment 1.10: “It would be helpful to add a brief discussion of how climate change (mainly increasing temperatures) would impact emissions (e.g., biogenics, exhaust)”

Response 1.10: After the statement on (current) L139-140:

“The meteorology for calendar year 2016 is used for all scenarios so that the projected air quality changes can be attributed solely to emissions changes.”

We have added

“Note that, under higher temperatures in a changing climate, vehicle exhaust emissions and biogenic (e.g., isoprene) emissions may increase, which could be explored in future work.”

Comment 1.11: “I appreciate the full Table of model performance metrics, and although all criteria are within acceptable limits, I’m curious how the model performs by region. One suggestion would be to include ozone and PM2.5 validation at the specific sites/monitors mentioned in the text,”

Response 1.11: Our manuscript follows the established guidance by the U.S. Environmental Protection Agency for model performance contained in the reference: *U.S. Environmental Protection Agency, “Modeling Guidance for Demonstrating Air Quality Goals for Ozone, PM2.5, and Regional Haze, EPA 454/R-18-009,” 2018.*

It is important to note that the model performance is evaluated at a seasonal/annual scale and separated between primarily urban and primarily rural measuring networks. The results of the model are not used in an absolute sense, but rather in a relative sense so that the model “response” is used to scale current design values to their future projections. This alleviates concerns over site-specific performance.

Comment 1.12: “I think the main text could use a bit more discussion of VOC emissions changes”

Response 1.12: Due to space constraints, we have added the following paragraph to Supplementary Note 2 after Figure S5:

“VOC emissions are largely driven by changes in four source categories: on-road vehicles, non-road sources, oil and gas activity, and other nonpoint sources. VOC emissions from on-road vehicles and non-road sources decline due to fleet turnover in the 2035 Limited Electrification scenario, but they are offset by emission increases from oil and gas activity and other nonpoint sources, such as solvent utilization and commercial and residential fuel combustion. High Electrification without Carbon Price can further reduce VOC emissions from on-road and non-road mobile sources due to further levels of electrification. VOC emissions from oil and gas sources decline in the 2035 High Electrification with Carbon Price scenario due to the reduced activity in this sector.”

Comment 1.13: “Since it always seems to be a caveat picked up on, how are manufacturing (battery) emissions included?”

Response 1.13: Battery manufacturing is not represented as a separate activity in the model. However, emissions associated with the manufacturing of vehicles are likely to be second order in terms of magnitude relative to emissions related to vehicle fuel use.

Comment 1.14: “Can you please provide more details on the WRF simulation performance and how that may have impacted your simulations? (the link provided in the reference is not functional)”

Response 1.14: We updated the reference for EPA’s evaluation of their WRF simulation to: https://www.epa.gov/sites/default/files/2020-10/documents/met_model_performance-2016_wrf.pdf (accessed 27 June 2022).

In the SI, after the statement “This EPA WRF dataset is used in this study,” we have added:

“EPA evaluated surface winds, temperature, water vapor, and precipitation patterns against observations and found that overall performance was adequate although spatial and temporal variations in performance were noted. Using 12-km grid resolution tended to improve WRF performance over using 36-km grid resolution, and there was a tendency for better WRF performance in the Eastern U.S. than the Western U.S., which can be attributed to the complex terrain of the West.”

Comment 1.15: “Given the offline nature of WRF-CAMx, a brief discussion of the uncertainty in including aerosol feedbacks might be helpful.”

Response 1.15: We added the following statement in the SI after the paragraph added above: “Running WRF and CAMx independently is efficient, because WRF is run only once, but has the limitation of omitting aerosol feedbacks to WRF meteorology although aerosol feedbacks to photolysis reactions are included within CAMx. Uncertainties in PM/O₃ responses introduced by omitting this feedback are likely much smaller than other factors such as the magnitude of the emission changes.”

Comment 1.16: “Given the proprietary nature of REGEN, it will be difficult to repeat this type of analysis, will the emissions be available?”

Response 1.16: We added a Data Availability statement at the end of the paper: “Source data underlying all figures and all other non-proprietary data supporting this study are available from the corresponding author upon reasonable request.”

Comment 1.17: “Given the population density and fleet of metro regions, a few more statistics on these regions might be warranted.”

Response 1.17: We have added five additional $PM_{2.5}$ time series, one from each domain, in Supplementary Note 4.

Reviewer #2

Comment 2.1: “This paper proposes a combined modelling approach to understanding the impact that energy system electrification can have on AQ in contiguous United States. The authors combine an energy system model, including a dispatch and capacity planning model linked to a demand model, with a full-form photochemical air quality model of the contiguous United States.

The paper is well written, relatively clear, and well structured. The approach is designed to transfer emissions trajectories from the energy model results to a full AQ model to allow the modellers to say something about the local concentrations of criteria pollutants and understand how different scenarios improve or worsen air quality. This has an important future link to human health (not addressed) which is mentioned, and represents an approach that I have not seen before. It also responds to the issue that regional energy models are often unable to say anything about the impact of their emissions on local AQ because of the lack of granularity and the very local nature of AQ impacts of emissions. Setting up such a methodology - or the first steps of it - is important work.”

Response 2.1: Thank you for your encouraging comments and constructive feedback, which have been helpful as we refined the manuscript.

Comment 2.2: “I will caveat my statements by the fact that my modelling expertise is restricted to energy system modelling. As such my comments will not touch on the AQ modelling approach or assess their suitability.

Noteworthy results:

Results suggest the following:

- *That electrification and decarbonisation policies have the potential to significantly improve AQ.*
- *That this improvements has limits linked to what the source of the AQ emissions are and that tighter carbon constraints for example may not lead to further improvements in AQ beyond a certain plateau.*
- *That methods that apply constant emissions estimates will tend to under-estimate the emissions reduction potential under different decarbonisation scenarios.*
- *That emissions reductions and AQ improvements are possible without decarbonisation policy - i.e. under optimistic assumptions about advanced end-use technologies.*

These results are sound and make sense in line with what other energy system modelling results would suggest. The dynamics that the scenarios highlight here are not surprising or novel from an energy system modelling perspective. Equally, while not an expert, I feel that the operation of the AQ model in itself yields expected dynamics in relation to pollutant circulation, secondary pollutant formation, etc. I believe that novelty in this work lies in the linking of the models and the ability to complete the picture of the energy model results with a detailed translation of their local impact. I would therefore re-focus part of the discussion to make this much plainer to the reader.”

Response 2.2: This is a great summary of the key takeaways from our paper. To emphasize the novelty of our paper, we emphasize the linkage in a few places:

- Abstract: “In this study, we evaluate the CO₂ and air quality co-benefits of electrification scenarios by employing a novel linking of a detailed energy systems model and a full-form photochemical air quality model in the United States.”
- Introduction contribution paragraph (beginning with “This study distinguishes itself...”): “...our paper is the first to link a detailed energy systems model with a full-form photochemical air quality model of the contiguous United States to examine scenarios that present a range of potential futures with increasing degrees of electrification, evaluated with an economy-wide

model. Linking these models provides a more complete picture of energy system transformations and their localized air quality impacts.”

- Discussion: “While earlier energy systems decarbonization modeling is consistent with these results and the air quality insights also are directionally consistent, the linked nature of our modeling is a unique contribution of our study, which provides a more complete portrait of these linked transformations.”

Comment 2.3: “Similarly, on the notion that decarbonisation policies can support AQ emissions but that this will plateau - and results from the AQ modelling showing that different parts of the US will face different persistent problems that may not meet NAAQS. Could the authors consider making recommendations or comments on the energy policies that could, if implemented in chosen areas, be shown to tackle this problem? (I realise this is partially alluded to with the paragraph p13 line 380 - but this only states what is missing in terms of existing policy rather than venturing a suggestion as to what types of policies in what areas and over what time-frames could be tested to remedy these tougher AQ emission issues.)”

Response 2.3: We added text to the Discussion section on possible policies and incentives to achieve air quality goals: “While the results of this study demonstrate the significant co-benefits of end-use electrification and electric sector decarbonization for local and regional air quality, and thus human and ecosystem health, some outstanding questions remain. First, it is quite possible that U.S. regulated air pollutant emission standards will be further tightened between now and 2050. In that case, emissions reductions from electrification may not be enough to drive air pollutant concentrations down to the newly required levels everywhere. That would imply additional measures being needed for non-electrified end-uses and non-combustion sources: e.g., improvements in combustion-based emission control technologies (whether fossil or biomass); development of lower-volatility, less reactive solvents; best management practices for agricultural emissions controls; and advancements in reducing primary PM_{2.5} emissions from industry and fugitive dust from roads and other non-combustion sources. Policies and incentives to achieve these goals could include strengthening regulatory approaches at the federal level under existing Clean Air Act authorities (e.g., EPA’s Cross-State Air Pollution Rule to assist downwind states to attain fine particle or ozone NAAQS), tax credits for lower-emitting technologies, and state-level ambient air quality standards. Alternatively, it could suggest that further electrification and accelerated timeframes will be needed to achieve more stringent air quality standards over the long term. Whether electrification would be a more cost-effective alternative at the margin than end-of-pipe or other emissions controls in a future scenario is an open question and would require further study.

Second, while this study assumes a quickly growing economy-wide carbon price in several of its scenarios, it does not explicitly incorporate some of the latest U.S. federal policy proposals for achieving net-zero electric sector by 2035 and economy-wide CO₂ emissions in the years post-2035 (e.g., clean electricity standards, vehicle and building appliance mandates, updates to the official social cost of carbon estimates used for rulemaking, etc.). However, reaching net-zero in the electric sector and across the economy may be achieved using a range of approaches that have very different implications for air quality [53]”.

Comment 2.4: “Having reviewed the US-REGEN model documentation (briefly!) and considered the paper and the SI - I'm afraid I feel that the information provided on the energy model and the scenarios that are presented here is insufficient.”

Response 2.4: In addition to adding detail on the model and scenarios in responding to other specific comments from the two reviewers, we added Table 1 to the Methods section with additional descriptions of key model features, assumptions, specific sections of the documentation corresponding to key assumptions, and resources for additional information.

Comment 2.5: “Scenarios:

1) While the description of the scenarios is clear enough to give the reader an understanding of how they differ, their key assumptions are not listed. This makes it difficult to i) understand in absolute terms what "more optimistic assumptions" or "accelerating the cost and performance assumptions" means and whether this is realistic. I would invite the authors to provide additional SI information on the key assumptions that differentiate the scenarios one from the other.”

Response 2.5: We added detailed descriptions of the scenario assumptions in Supplementary Note 6, which also adds figures to illustrate key assumptions. We also added a sentence to the main text and Methods that: “Detailed scenario assumptions are provided in Supplementary Note 6.”

Comment 2.6: “2) Carbon price data is explicit for the third case, but is not "justified" or put into context: 271\$ is a very specific number, where does this come from and why? Also, how does this compare to future cost of carbon projections in other US scenario exercises?”

Response 2.6: We added text to the Methods section to justify the carbon price trajectory and place it in context of recent U.S. scenario exercises: “The starting price in 2025 and adjustment over time are similar to the proposal by the Climate Leadership Council in 2019, the border carbon adjustment proposed by Senator Whitehouse in 2022, and multi-model scenarios for the U.S. [61, 43], including a recent study of the 2030 U.S. climate target that estimates a marginal cost of CO₂ reductions between \$36 to \$155/t-CO₂ by 2030 [43].”

Comment 2.7: “3) What constraints climate policies - if any - do these scenarios all include? This relates to my question about e.g. the US NDC which appears lower down.”

Response 2.7: We added a list of policies (including climate policies) represented in all scenarios at the beginning of the “Scenarios” section: “All scenarios include representations of significant on-the-books federal and state policies and incentives as of June 2021, including:

- State-level renewable portfolio standards, including technology-specific carveouts for solar
- State-level clean electricity standards with state-specific definitions of qualifying resources
- State-level offshore wind mandates (CT, MA, MD, ME, NJ, NY, RI, VA) and energy storage mandates (CA, NJ, NY, VA)
- California AB32, represented as a carbon tax based on projections by the California Air Resources Board
- Regional Greenhouse Gas Initiative (RGGI) cap-and-trade system
- Current Clean Air Act Section 111(b) new source performance standards
- Investment tax credits and production tax credits

The carbon tax scenario layers this economy-wide tax on top of these existing policies. Note that scenarios do not explicitly include the updated U.S. Nationally Determined Contribution pledge to reduce emissions by 50-52% by 2030 from 2005 levels, since formal policies are not yet in place to achieve this target, though the High Electrification with Carbon Price scenario is consistent with this target (Fig. S15). Scenarios also incorporate announced retirement dates for coal plants.”

Comment 2.8: “Energy model:

1) Understanding what core data underpins / is found in all scenarios and is important for the results is not easy. Reviewing the model documentation (which is extensive - and beyond much of the documentation that is available for other modelling frameworks!) there are tables of costs and efficiencies, yes, but there are also different sets of assumptions for different technology options - and the potential to include technologies or not. Along with the SI information for the scenario points it would

therefore be welcome to have, at least, a clear statement of what is included in the model with direct reference to relevant parts of the documentation - both in terms of technology options (including e.g. CCS and DAC for example) and in terms of data used. I understand this is a large undertaking and would consider the authors' selection of what constitutes critical information acceptable.

2) One example of this is - what temporal resolution is US-REGEN run on for this work?"

Response 2.8: We added Table 1 to the Methods section with additional descriptions of key model features, assumptions, and resources for additional information (including a row for the temporal resolution). The column on documentation links to specific sections of the US-REGEN documentation for readers who are interested in learning more. All of the questions mentioned by the reader are included as rows in the table.

Comment 2.9: "Methodology

1) My main comment on the methodology pertains to the description and language used to describe the energy model. This language and description has a strong tendency to describe it as "detailed structural energy systems model", "integrated electric and energy system model", "detailed energy systems model", "uniquely capable" - the list goes on and leads the reader to assume it is a whole energy system model. US-REGEN is a capacity planning and dispatch model of the ELECTRICITY system, combined with a logit model of demand side service delivery. The entire upstream and process sector producing any other commodity but electricity (and H2 I believe), is missing. This has an important impact on the interpretation of the results - particularly since a Logit model will be parameterised against existing macro-economic relationships. This to me creates two issues that are not highlighted clearly, first that a large part of the whole energy system is not endogenised, and second that the demand side model in US-REGEN will to an extent apply our current expectations of the energy system to determine what should be installed or not. I completely recognise that different modelling approaches exist, and I believe this one has value. However its characteristics need to be made much clearer throughout the paper and the SI."

Response 2.9: With regard to the issue that the upstream and process sector is not “endogenized,” it is true that in the version of US-REGEN used in this study the supply of non-electric fuels is limited to conventional fossil fuels characterized by fixed technologies (e.g., petroleum refining). However, these activities are included, just in a less detailed fashion, so that the model does in fact cover the whole of the energy system. For the purpose of this study, we believe the model is appropriately capturing structural change across the energy system associated with increased electrification. We have articulated this more clearly in the model description section of the SM. As an aside, subsequent research has expanded the representation of non-electric fuel supply in US-REGEN to include a broader set of technologies and options, including a range of low-carbon fuel pathways, which will be used for future research on the air quality implications of economy-wide decarbonization pathways.

With regard to the logit model parameterization, please see the response to Comment 2.10.

Comment 2.10: "2) Extending this - please comment on how the logit model is parameterised for demand side technologies that are not yet available or do not yet have extensive data available with a potential reference to the section of the US-REGEN documentation that details this."

Response 2.10: There are two components of the end-use model that use a logit formulation: The buildings sectors (specifically, space and water heating and other dual-fuel appliances) and personal light-duty vehicle sectors. In these sectors, the logit formulation is intended to represent behavioral heterogeneity rather than pure optimization. The market allocation in the logit model is still driven by assumed costs: The least-cost option does not capture 100% of the market allocation, but adoption rates are greater for technologies with greater cost advantage. The parameterization of the logit model

determines the sensitivity of the market allocation (within a given nest in the nested structure) to assumed (“observed”) costs, or conversely the influence of non-modeled (“non-observed”) or costs or preferences, which by construction are assumed to be independently and identically distributed for each technology in the nest. In the case of the buildings sectors, the model is primarily evaluating trade-offs between electric heat pumps, electric resistance, and gas or other non-electric technologies. While the cost and performance of these technologies may evolve over time, they all exist today and have observable market shares that vary across regions of the U.S. and segments of the building stock. Thus, the choice of logit model parameters, which again are not associated with technologies per se but rather with nests defining groups of technologies among which market share is allocated, can be informed by currently observed patterns. Nonetheless, improvements in heat pumps over time increase the cost advantage and drive additional adoption relative to today’s market shares.

In the case of light-duty vehicles, until very recently there has been essentially only one technology deployed, namely the ICEV. With the rapid development and commercialization of PHEVs and EVs over the past several years, adoption rates are growing, but current market shares cannot be construed as representing an equilibrium allocation given the very early stage of development and remaining barriers to scale. For this reason, the logit parameters in this light-duty vehicle sector are not informed by current data. Rather they are chosen to represent a hypothetical long-run equilibrium. Moreover, the model includes a lagged function assigned market allocation in a given time step as a linear combination of the logit model’s allocation based on current costs and the previous time steps actual allocation. Thus, there is a gradual convergence to the outcomes of the logit formulation when those outcomes are significantly different than the current outcome. In practice, the modeled total costs of electric vehicles suggest a significant cost advantage over ICEVs even in the near-term, with the advantage increasing over time as battery costs fall further and potential policy incentives increase the relative costs of non-electric fuels. The result is increasing adoption based on economic fundamentals, the pace of which is not strongly impacted by the choice of logit parameters. Thus, we believe it is not the case that the model is inappropriately anchored to current system characteristics. We have included a few sentences summarizing the key structural elements of the end-use model in the model description section of the SM. More information about the formulation and the parameter choices is available in the model documentation in the relevant sections of Chapter 3.

Comment 2.11: “3) The baseline scenario is chosen is hypothetical (admittedly, as all baselines and scenarios are), but seems to depart from “current trends”. It is not clear in the paper how this reference scenario compares to e.g. current roll-out trends for Evs which here are “limited”, or, “imposing a constraint on the electrification of buildings so that shares do not increase beyond today’s levels”. Could you please provide a paragraph / explanation as to how this future compares to what is currently happening in the US energy system? And an explanation where there is a cutoff (e.g. the case of the buildings) as to why this is appropriate for the modelling exercise?”

Response 2.11: We added a sentence to the initial scenario description to clarify for the Limited Electrification scenario that: “Comparing this hypothetical benchmark scenario with others can quantify the impact of electrification on outcomes of interest.” We also added a sentence to the description of the High Electrification case that: “This scenario more closely approximates expected market trends than does the Limited Electrification case, which allows us to evaluate the incremental effects of such electrification.” In this paper, we are interested in the effects of electrification on air quality, so we need a counterfactual case with limited electrification. We could have had a baseline/control case with “reference” electrification (i.e., with current market trends) and a treatment case with even higher electrification, but that would only show the effects of that incremental electrification.

To illustrate how these scenarios compare with current market trends, we added SI Fig. 16 to Supplementary Note 5. This figure illustrates electric vehicle deployment in the three scenarios and

compares these results with trends from a recent multi-model study. We also added a sentence in the main text in the first paragraph of the “Energy System and Emissions Results” section that: “Note that the levels of electric vehicle deployment in the High Electrification scenario more closely resemble anticipated electrification based on projected market trends (SI Fig. S16).”

Comment 2.12: “4) It is not clear why the results from the modelling presented for the different scenarios do not all include 2050. Could you please consider including additional scenario result outputs for the energy system model for 2050 for the limited electrification and high electrification scenarios - or alternately justifying why the choice has been made to present results to 2050 only for the carbon price scenario.”

Response 2.12: All energy system scenarios in the REGEN model were run in five-year intervals through 2050. However, due to the computational and resource intensity of runs, air quality scenarios in CAMx are only run for select years and scenarios of interest. We focused on performing the suite of 2035 runs since this allowed us to explore the impact of electrification over a time horizon that allowed for significant penetration of technologies while at the same time being informative to the deadlines of upcoming State Implementation Plans that would be necessary for attainment demonstration of potential revisions to the National Ambient Air Quality Standards for ozone and PM_{2.5}. These runs were able to demonstrate differences over time frame across key scenarios. As a “add-on” to the project, we felt that we could explore the potential of further air quality improvements due to electrification into 2050 by extending most aggressive scenario and thereby provide insight as to potential floor of concentrations of the key air pollutants via electrification.

Comment 2.13: ““Insights gained from detailed modelling”

This section discusses the difference in emissions reductions expected if analysis either endogenises electricity emission factors or instead uses an approach that applies fixed emission factors (with different fixing methods). This is interesting and useful. It is however important to note that IAM or whole energy system modelling approaches typically endogenise carbon content of electricity calculations as standard. This therefore - in and of itself - does not represent a step up as compared to other comparable energy system modelling approaches. This does not remove the fact that any analysis that uses fixed emission factors for this will underestimate emissions savings as the electricity sector decarbonises - but is rather a statement that this is an expected result. This section may be more impactful if it stated this more clearly and focused more on the analytical approaches that do use fixed emission factors and are typically used for policy.”

Response 2.13: We altered the wording at the beginning of this section to make the clarifications mentioned by the reviewer. We added the sentence: “These findings contrast with earlier studies suggesting more limited emissions effects of electrification, especially those that use short-run marginal emissions estimates [20, 10], which characterize marginal emissions from fixed electricity systems and do not account for structural changes over time as many energy systems models do.” For the first bullet, we added: “Many systems modeling methods endogenize CO₂ emissions from the power sector and implicitly use this approach.”

Comment 2.14: “Useful additional information

• Consider including data for US emissions breakdown and their trends, and energy consumption by sector and fuel in the SI for reference and comparison.”

Response 2.14: We added Figure S19 to show U.S. emissions broken out by sector and compared this with emissions projections across the three core scenarios in the paper. Figure S17 illustrates energy consumption by fuel and end-use historically and across different scenarios in the study.

Comment 2.15: “• Please include commentary as to how the scenarios you run compare to / meet current US energy and emissions targets. I understand that different states will have different levels of ambition and that this could be difficult to present, but an understanding of how the scenarios measure up to e.g. US NDC ambitions would help to place the results in context.”

Response 2.15: We added Fig. S19 to Supplementary Note 5 to illustrate the CO₂ trajectories in our scenarios relative to historical trends and the updated U.S. NDC. We added text to the main paper in the “Energy System and Emissions Results” section to describe the CO₂ relative to these goals: “Economy-wide CO₂ emissions decline across all scenarios (Figure 1, bottom), though the rate and extent vary by scenario. CO₂ reductions relative to 2005 levels are 33% in the Limited Electrification scenario, 44% in the High Electrification, and 78% in the High Electrification with Carbon Price scenario. In April 2021, the U.S. updated its pledge as part of the Paris Agreement to reduce emissions by 50-52% by 2030 from 2005 levels [43]. As shown in Fig. S15, the High Electrification with Carbon Price scenario is consistent with this target, while the other scenarios fall short, entailing CO₂ declines by 19-28% by 2030.”

Comment 2.16: “P2 line 33 - other methods for decarbonising energy systems include lowering demand - see (Grubler et al, 2018).”

Response 2.16: We revised this sentence to: “Reducing power sector emissions and then using that low-emitting electricity to reduce end-use emissions is employed alongside energy efficiency, carbon capture and removal, other low-carbon fuels, and demand-side responses to decarbonize energy systems and address climate change.”

Comment 2.17: “P2 line 68 "annual snapshots" is not appropriate. While IAMs do have a coarser temporal resolution, they will typically represent seasons X day types X day parts in some detail. They do not present a "single year approach" please amend or clarify.”

Response 2.17: We modified this sentence to read: “IAMs themselves and other energy system models can have relatively coarse temporal [26] and spatial resolution [27, 28], and use more aggregated source sector (e.g., economic sector) than standard emission inventories used in AQM (e.g., Source Classification Code, SCC, level).”

Comment 2.18: “P2 line 42 ref [6] it is notable that this does not include eco-system services, the valuation of which is starting to shift this picture and would be worth recognising.”

Response 2.18: We altered this sentence to: “Research suggests that monetized benefits of climate change mitigation are high for air quality improvements, which have immediate and localized benefits [6].” so that we are not explicitly making claims about the relative value of air quality benefits vis-à-vis climate damages, ecosystem services, and other potential co-benefits of mitigation. Ultimately, we are simply establishing as context that lowering CO₂ has sizeable benefits due to air quality improvements.

Comment 2.19: “P3 line 76 "range of scenario specific drivers" please detail where appropriate.”

Response 2.19: We expanded this sentence to read: “These features provide endogenous responsiveness of electricity demand and hourly electric load shapes to a range of scenario-specific drivers including incentives for electrification and CO₂ policy, which improves upon previous efforts with exogenous emissions scenarios, and end-use representations, and sectoral adoption sensitivities [32, 37, 6, 38].”

Comment 2.20: “P3 line 85 "established methods for assssing future AQ" please reference”

Response 2.20: We added a citation to the following document from the US EPA: U.S. Environmental Protection Agency, “Modeling Guidance for Demonstrating Air Quality Goals for Ozone, PM2.5, and Regional Haze, EPA 454/R-18-009,” 2018.

Comment 2.21: “P3 line 105 “behavioral” please qualify - this is based on an economic understanding of behaviour rather than e.g. on the parameterisation of an agent based model with different actors represented using observational data.”

Response 2.21: We added text to the Methods section to clarify: “Behavioral features of energy consumers are represented by incorporating disutility costs associated with purchase decisions such as refueling station availability for passenger vehicles [58], using a simulation-based approach to capture unobserved preference heterogeneity across households (Table 1), and explicitly breaking out structural classes to capture observed heterogeneity across households and firms. Note that this framework does not capture possible interactions of heterogeneous agents, where adoption and utilization decisions are influenced by the choices of others, as many agent-based models do [59].”

Comment 2.22: “P3 line 109: “rather than relying on a historical snapshot” - please include a reference clearly to studies that do this and clarify in text. Dynamic marginal emissions related to electricity production are typically standard in most energy system modelling exercises.”

Response 2.22: We expanded this sentence and added a reference: “Moreover, the integrated representation of electricity supply and demand provides a dynamic and scenario-consistent treatment of the marginal emissions from increased electric generation to support electrification, taking into account structural changes to the generation mix over time rather than relying on a historical snapshot, as many marginal emissions studies of electrification do [20].”

Comment 2.23: “P4 line 157 “Unlike [...]scenarios” could you please explain why this is the case? What is it about these runs where - as I understand it - no carbon constraints are implemented that is different from these other exercises where coal is maintained/increased?”

Response 2.23: We expanded this sentence: “Unlike other studies, higher electrification does not lead to increases in dispatch from existing coal in these scenarios even without a decarbonization policy, as the endogenous investment and retirements lead to declining coal generation, which contrasts with dispatch-only short-run marginal emissions studies [20].”

Comment 2.24: “P13 line 367 paragraph: could the authors consider discussing the residual non-electricity fuel consumption in different sectors to highlight areas where this “residual emissions” problem is expected?”

Response 2.24: We added a sentence on residual emission to this paragraph: “The bottom panel of Figure 1 illustrates how residual emissions are highest for sectors with higher marginal abatement costs, including non-light-duty transport (e.g., aviation, shipping), high-temperature heat for industry, and sectoral options on the steep portion of sectoral abatement cost curves (e.g., space heating for buildings in the coldest climates), which aligns with the broader decarbonization literature [52].”

Comment 2.25: “P14 line 407 reference [50] search engine results for this send the reader to publications from 2012 - consider adding a link in the reference as I assume this should point to the us-regen-docs.epri.com reference lower down on line 423.”

Response 2.25: We updated this sentence as follows: “Cost and performance estimates for different supply-side technologies come from the literature, expert elicitations, and EPRI’s Technology

Assessment Guide, which are provided in the US-REGEN documentation available at: <https://us-regen-docs.epri.com>.”

REVIEWERS' COMMENTS

Reviewer #1 (Remarks to the Author):

The authors have addressed my concerns and I have no further recommended edits prior to publication.

Reviewer #2 (Remarks to the Author):

The authors have made a real effort to consider and respond to my initial comments on this paper. This is much appreciated - thank you.

Considering that most responses are close to being agreeable and that the general objective of the comments in the first place was to invite further detail rather than to question the fundamental premise or approach of the paper - my second review is short and responds to the numbered responses from the authors.

I realise that some of these entail additional text in the main manuscript which is supposed to be limited in length, however some of the clarifications would help the reader know what they are reading without the need to swap to a lengthy SM document.

2.1 - I hope they were truly helpful rather than irritating.

2.2 - Appropriate

2.3 - Appropriate

2.4 - While reading through this again I note that non GHG emission results from US REGEN are split using other emissions data to produce the input to the AQ modelling. This is clear in the SM but deserves a sentence in the main manuscript.

2.5 - Appropriate

2.6 - You choose a 7% increase p.a. and a seed value of \$50/t, but you reference the Climate Leadership Council 2019 who suggest a 5% p.a. growth rate and a \$40/t seed value. Could you be just a little clearer on the choice - i.e. modify the sentence so that it reads something like "While the Climate Leadership Council 2019 suggest xxxxx, this study applies yyyyyy in reference to REF and REF who suggest zzzzz". At the moment the references are in your response but the reasoning is not.

2.7 - Appropriate

2.8 - Appropriate

2.9 - Having reviewed the answers to this and to other comments, I would revise my comment here - moving away from the endogenisation criticism towards one of semantics. Your study does capture the appropriate system for the purpose of reviewing the impact of electrification. The question then becomes one of what the reader expects and understands based on certain terminology being employed. My background is in whole systems energy modelling - and any reference to a variation of that terminology leads me to expect a fully integrated model, that represents full supply chains and interconnected sectors and trades off commodity import, transformation, dispatch, transport and use across the entire energy system of the region. Your response here confirms that this is not what your approach does. While I realise that there are no hard and fast rules about semantics, could I nevertheless invite you to adjust the wording that you use, or, include an explicit disclaimer so that the reader knows what you mean when you use these wordings.

2.10 - Please consider including the type of explanation offered in the response in relation to the different parameterisation approaches for the logit models in the main body of the paper - or - depending the explanation in the SM.

2.11 - I would suggest making the choice that is made here plainer to the reader by replacing "Comparing this hypothetical benchmark scenario with others can quantify the impact of electrification on outcomes of interest." with "This hypothetical benchmark was constructed to display conservative levels of electrification that, when compared to results from other scenarios, helps to better quantify the impact of electrification on outcomes of interest.". All other changes are good and should also be kept.

2.12 - This makes sense - could you add a sentence to the paper to this effect.

2.13 - Suggest the following change: "Many systems modeling methods endogenize CO2 emissions from the power, and all other, sectors and implicitly use this approach."

2.14 - Appropriate

2.15 - Appropriate (note typo in reviewer response S15 but main paper edits seem correct)

2.16 - Suggest the following change: replace "and demand side response" with "the lowering of energy demand through structural and societal change, as well as demand side response technologies, "

2.17 - 2.25 Appropriate.

I do not think that any of these edits represent serious issues or impediments to publication - they simply ask for a little more detail and specificity in some of the statements made in the main manuscript.

Thank you for all the effort you've already put into what is an impressive piece of work.

Response to Reviewers

Reviewer #2

Comment 2.0: "The authors have made a real effort to consider and respond to my initial comments on this paper. This is much appreciated - thank you.

Considering that most responses are close to being agreeable and that the general objective of the comments in the first place was to invite further detail rather than to question the fundamental premise or approach of the paper - my second review is short and responds to the numbered responses from the authors.

I realise that some of these entail additional text in the main manuscript which is supposed to be limited in length, however some of the clarifications would help the reader know what they are reading without the need to swap to a lengthy SM document."

Response 2.0: Thanks again for your constructive comments.

Comment 2.4: "While reading through this again I note that non GHG emission results from US REGEN are split using other emissions data to produce the input to the AQ modelling. This is clear in the SM but deserves a sentence in the main manuscript."

Response 2.4: We added a couple sentences in the "Modeling Energy System and Air Quality Impacts of Electrification" to summarize: "These emissions are processed to develop more temporally, spatially, and chemically refined inputs for air quality modeling. US-REGEN emissions by sector, source, fuel type, and region are cross-referenced to Source Classification Codes (SCC), county, and North American Industry Classification System (NAICS) code (see Supplementary Note 2 for more information)."

Comment 2.6: "You choose a 7% increase p.a. and a seed value of \$50/t, but you reference the Climate Leadership Council 2019 who suggest a 5% p.a. growth rate and a \$40/t seed value. Could you be just a little clearer on the choice - i.e. modify the sentence so that it reads something like "While the Climate Leadership Council 2019 suggest xxxxx, this study applies yyyyyy in reference to REF and REF who suggest zzzzz". At the moment the references are in your response but the reasoning is not."

Response 2.6: We modified this sentence at the end of the Methods section to clarify: "The starting price in 2025 is comparable to the proposal by the Climate Leadership Council in 2019 (which was proposed to start at \$40/tCO₂ in 2017 USD by 2021 and then increase annually at 5% above inflation [60]) and multi-model scenarios for the U.S. [61, 43], including a recent study of the 2030 U.S. climate target that estimates a marginal cost of CO₂ reductions between \$36 to \$155/t-CO₂ by 2030 [43]."

Comment 2.9: "Having reviewed the answers to this and to other comments, I would revise my comment here - moving away from the endogenisation criticism towards one of semantics. Your study does capture the appropriate system for the purpose of reviewing the impact of electrification. The question then becomes one of what the reader expects and understands based on certain terminology being employed. My background is in whole systems energy modelling - and any reference to a variation of that terminology leads me to expect a fully integrated model, that represents full supply chains and interconnected sectors and trades off commodity import, transformation, dispatch, transport and use across the entire energy system of the region. Your response here confirms that this is not what your approach does. While I realise that there are no hard and fast rules about semantics, could I nevertheless invite you to adjust the wording that you use, or, include an explicit disclaimer so that the reader knows what you mean when you use these wordings."

Response 2.9: We added a sentence up front in the paper in the “Modeling Energy System and Air Quality Impacts of Electrification” to delineate features that fall outside of the scope of the modeling: “Note that the energy system model scope for this analysis does not include endogenous representations of fossil fuel prices, biofuel production, non-electric fuel movement, or general equilibrium effects.” We recognize that different readers will have different notions of what constitutes an “energy systems model,” so we hope that our descriptions throughout the paper help the reader to understand what is inside and outside of the scope of this particular analysis.

Comment 2.10: “Please consider including the type of explanation offered in the response in relation to the different parameterisation approaches for the logit models in the main body of the paper - or - deepening the explanation in the SM.”

Response 2.10: We added a couple sentences to the Methods section: “Parameter selection for the logit model is discussed in Supplementary Note 1. The parameterization of the logit model determines the sensitivity of the market allocation (within a given nest in the nested structure) to assumed (“observed”) costs, or conversely the influence of non-modeled (“non-observed”) or costs or preferences, which by construction are assumed to be independently and identically distributed for each technology in the nest.”

We also added more detail in Supplementary Note 1: “In these sectors, the logit formulation is intended to represent behavioral heterogeneity rather than pure optimization. The market allocation in the logit model is still driven by assumed costs: The least-cost option does not capture 100% of the market allocation, but adoption rates are greater for technologies with greater cost advantage. The parameterization of the logit model determines the sensitivity of the market allocation (within a given nest in the nested structure) to assumed (‘observed’) costs, or conversely the influence of non-modeled (‘non-observed’) or costs or preferences, which by construction are assumed to be independently and identically distributed for each technology in the nest. In the case of the buildings sectors, the model is primarily evaluating tradeoffs between electric heat pumps, electric resistance, and gas (or other non-electric) technologies. While the cost and performance of these technologies may evolve over time, they all exist today and have observable market shares that vary across regions of the U.S. and segments of the building stock. Thus, the choice of logit model parameters, which are not associated with technologies per se but rather with nests defining groups of technologies among which market share is allocated, can be informed by currently observed patterns.

In the case of light-duty vehicles, until very recently there has been essentially one technology deployed, namely the internal combustion engine vehicle (ICEV). With the rapid development and commercialization of electric vehicles over the past several years, adoption rates are growing, but current market shares cannot be construed as representing an equilibrium allocation given the very early stage of development and remaining barriers to scale. For this reason, the logit parameters in this light-duty vehicle sector are not informed by current data but are chosen to represent a hypothetical long-run equilibrium. Moreover, the model includes a lagged function assigned market allocation in a given time step as a linear combination of the logit model’s allocation based on current costs and the previous time step’s actual allocation. Thus, there is a gradual convergence to the outcomes of the logit formulation when those outcomes are significantly different than the current outcome. In practice, the modeled total costs of electric vehicles suggest a significant cost advantage over ICEVs even in the near-term, with the advantage increasing over time as battery costs fall further and potential policy incentives increase the relative costs of non-electric fuels. The result is increasing adoption based on economic fundamentals, the pace of which is not strongly impacted by the choice of logit parameters. More information about the formulation and the parameter choices is available in Chapter 3 of the model documentation.”

Comment 2.11: "I would suggest making the choice that is made here plainer to the reader by replacing "Comparing this hypothetical benchmark scenario with others can quantify the impact of electrification on outcomes of interest." with "This hypothetical benchmark was constructed to display conservative levels of electrification that, when compared to results from other scenarios, helps to better quantify the impact of electrification on outcomes of interest.". All other changes are good and should also be kept."

Response 2.11: We made this suggested wording change.

Comment 2.12: "This makes sense - could you add a sentence to the paper to this effect."

Response 2.12: We added a paragraph to the Methods section: "All energy system scenarios in US-REGEN are run in five-year intervals through 2050. However, due to the computational and resource intensity of runs, air quality scenarios in CAMx are only run for select years and scenarios of interest. The analysis focuses on 2035 to explore the impact of electrification over a time horizon that allows for significant penetration of technologies while at the same time being informative to the deadlines of upcoming State Implementation Plans, which would be necessary for attainment demonstration of potential revisions to the NAAQS for ozone and PM_{2.5}."

Comment 2.13: "Suggest the following change: 'Many systems modeling methods endogenize CO2 emissions from the power, and all other, sectors and implicitly use this approach.'."

Response 2.13: We modified this sentence as follows: "Many systems modeling methods endogenize CO₂ emissions from the power sector (and all other sectors) and implicitly use this approach."

Comment 2.16: "Suggest the following change: replace 'and demand side response' with 'the lowering of energy demand through structural and societal change, as well as demand side response technologies,'."

Response 2.16: We revised the sentence as follows: "Reducing power sector emissions and then using that low-emitting electricity to reduce end-use emissions is employed alongside energy efficiency, carbon capture and removal, other low-carbon fuels, and demand-side responses (including the lowering of energy demand through structural and societal change) to decarbonize energy systems and address climate change."